# How hard is to distinguish graphs
# with graph neural networks?

**Andreas Loukas**
École Polytechnique Fédérale Lausanne
`andreas.loukas@epfl.ch`

## Abstract

A hallmark of graph neural networks is their ability to distinguish the isomorphism class of their inputs. This study derives hardness results for the classification variant of graph isomorphism in the message-passing model (MPNN). MPNN encompasses the majority of graph neural networks used today and is universal when nodes are given unique features. The analysis relies on the introduced measure of *communication capacity*. Capacity measures how much information the nodes of a network can exchange during the forward pass and depends on the depth, message-size, global state, and width of the architecture. It is shown that the capacity of MPNN needs to grow linearly with the number of nodes so that a network can distinguish trees and quadratically for general connected graphs. The derived bounds concern both worst- and average-case behavior and apply to networks with/without unique features and adaptive architecture—they are also up to two orders of magnitude tighter than those given by simpler arguments. An empirical study involving 12 graph classification tasks and 420 networks reveals strong alignment between actual performance and theoretical predictions.

## 1 Introduction

A fundamental goal in the analysis of graph neural networks is to determine under what conditions current networks can (or perhaps cannot) distinguish between different graphs [1–6]. The most intensely studied model in the literature has been that of message-passing neural networks (MPNN). Since its inception by Scarselli et al. [7], MPNN has been extended to include edge [8] and global features [9]. The model also encompasses many of the popular graph neural network architectures used today [10, 3, 11–16].

Roughly two types of analyses of MPNN may be distinguished. The first bound the expressive power of *anonymous* networks, i.e., those in which nodes do not have any access to node features (also known as labels or attributes) and that are permutation equivariant by design. Xu et al. [3] and Morris et al. [4] established the equivalence of anonymous MPNN to the 1st-order Weisfeiler-Lehman (1-WL) graph isomorphism test. A consequence of this connection is that anonymous MPNN cannot distinguish between regular graphs with the same number of nodes, but can recognize trees as long as the MPNN depth exceeds the tree diameter. Other notable findings include the observation that MPNN cannot count simple subgraphs [6], as well as the analysis of the power of particular architectures to compute graph properties [17, 18] and to distinguish graphons [19]—see also [20, 21, 5].

The aforementioned insights can be pessimistic in the *non-anonymous* case, where permutation equivariance is either learned from data [22, 23] or obtained by design [24]. With node features acting as identifiers, MPNN were shown to become universal *in the limit* [23], which implies that they can solve the graph isomorphism testing problem if their size is allowed to depend exponentially on the number of nodes [5]. The node features, for instance, may correspond to a one-hot encoding [10, 25, 22] or a random coloring [26, 27].

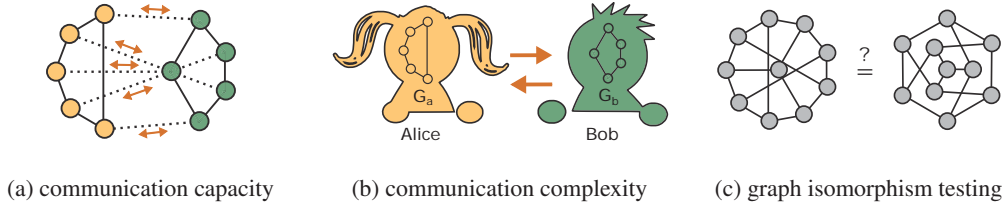

| (a) communication capacity | (b) communication complexity | (c) graph isomorphism testing |

Figure 1: (a) In MPNN, nodes exchange information by sending and receiving messages along edges. Communication capacity is the maximal amount of information that can be sent across two subgraphs (depicted in orange and green) (b) Communication complexity is the minimal amount of information needed so that two parties jointly compute a function $f$. (c) To determine whether graphs $G$ and $G'$ are isomorphic one may use an MPNN $g$ to test whether $g(G) = g(G')$.

At the same time, universality statements carry little insight about the power of practical networks, as they only account for behaviors that occur in the limit. Along those lines, recent work provided evidence that the power of MPNN grows as a function of depth and width for certain graph problems [23], showing that (both anonymous and non-anonymous) MPNN cannot solve many tasks when the product of their depth and width does not exceed a polynomial of the number of nodes. Nevertheless, it remains an open question whether similar results hold also for problems relating to the capacity of MPNN to distinguish graphs. Even further, it is unclear whether depth and width needs to grow with the number of nodes solely in the worst-case (as proven in [23]) or with certain probability over the input distribution.

## 1.1 Communication capacity and its consequences to distinguishing graphs

Aiming to study the power of MPNN of practical size to distinguish graphs, this paper defines and characterizes *communication capacity*, a measure of the amount of information that the nodes of the network can exchange during the forward pass (see Figure 1a). In Section 2 it is shown that the capacity of MPNN depends on the network's depth, width, and message-size, as well as on the cut-structure of the input graph. Communication capacity is an effective generalization of the previously considered product between depth and width [23], being able to consolidate more involved properties, as well as to characterize MPNN with global state [8, 9, 28] and adaptive architecture [29–32].

The paper then delves into the *communication complexity* of determining the graph isomorphism class. The theory of communication complexity compliments the definition of communication capacity as it provides a convenient mathematical framework to study how much information needs to be exchanged by parties that jointly compute a function [33] (see Figure 1b). In this setting, Section 3 derives hardness results for determining the isomorphism class of connected graphs and trees. It is shown that the communication capacity of any MPNN needs to grow at least linearly with the number of nodes so that the network can learn to distinguish trees, and quadratically to distinguish between connected graphs. The analysis stands out from previous relevant works that have studied subcases of isomorphism, such as subgraph freeness [34, 35] or those focused on anonymous networks [3–6, 17, 19]. In fact, the derived hardness results apply to *both* anonymous and non-anonymous MPNN and can be up to two orders of magnitude tighter than what can be deduced from simpler arguments. In addition, the proposed lower bounds rely on a new technique which renders them applicable not only to worst-case instances [23], but in expectation over the input distribution.

An empirical study reveals strong qualitative and quantitative agreement between the MPNN test accuracy and theoretical predictions. In the 12 graph isomorphism tasks considered, the performance of the 420 graph neural networks trained was found to depend strongly on their communication capacity. In addition, the proposed theory could consistently predict which networks would exhibit poor classification accuracy as a function of their capacity and the type of task in question.

## 2 Communication complexity for message-passing networks

Suppose that a learner is given a graph $G = (\mathcal{V}, \mathcal{E}, a)$ sampled from a distribution $\mathbb{D}$ that is supported over a finite universe of graphs $\mathcal{X}$. Throughout this paper, $\mathcal{V}$ will denote the set of nodes of cardinality $n$, $\mathcal{E}$ the set of edges, and $a$ encodes any node and edge features of interest. With $G$ as input, the

learner needs to predict the output of function $f : \mathcal{X} \to \mathcal{Y}$. This work focuses on graph classification, in which case $f$ assigns a class $y \in \mathcal{Y}$ (i.e., its isomorphism class) to each graph in the universe.

## 2.1 Message-passing neural networks (MPNN)

In MPNN, the node representation $x_i^{(\ell)}$ of every node $v_i \in \mathcal{V}$ is initialized to be equal to the node's attributes $x_i^{(0)} = a_i$ and is progressively updated by exchanging messages:

$$x_i^{(\ell)} = \text{UPDATE}_\ell\big(x_i^{(\ell-1)}, \; \{\text{msg}_{ij}^{(\ell)} \; : \; e_{ij} \in \mathcal{E}\}\big) \quad \text{for} \quad \ell = 1, \ldots, d,$$

where each message

$$\text{msg}_{ij}^{(\ell)} = \text{MESSAGE}_\ell\big(x_j^{(\ell-1)}, \; a_j, \; a_{ij}\big)$$

contains some information that is sent to from node $v_j$ to $v_i$.

Every neuron in a network utilizes some finite alphabet $\mathcal{S}$ containing $s = |\mathcal{S}|$ symbols to encode its state. For this reason, $x_i^{(\ell)}$ and $\text{msg}_{ij}^{(\ell)}$ are selected from $\mathcal{S}^{w_\ell}$ and $\mathcal{S}^{m_\ell}$, where $w_\ell$ and $m_\ell$ are the width (i.e., number of channels) and the message-size of the $\ell$-th layer. For instance, to represent whether a neuron is activated one uses binary symbols, whereas a practical implementation could use as symbols the set of numbers represented in floating-point arithmetic.

$\text{MESSAGE}_\ell$ and $\text{UPDATE}_\ell$ are layer-dependent functions whose parameters are selected based on some optimization procedure. It is common to parametrize these functions by feed-forward neural networks [7, 12, 9]. The rational is that, by the universal approximation theorem and its variants [36–38], these networks can approximate any smooth function that maps vectors onto vectors. If the network's output is required to be independent of the number of nodes, the output is recovered from the representations of the last layer by means of a readout function: $g(G) = \text{READOUT}\big(\{x_i^{(d)} : v_i \in \mathcal{V}\}\big)$. For simplicity, it is here assumed that no graph pooling is employed [39, 40], though the results may also be easily extended to account for coarsening [41–45].

**Global state.** In the description above, all message exchange needs to occur along graph edges. However, one may also easily incorporate a *global state* (or external memory) to the model above by instantiating a special node $v_0$ and extending the edge set to contain edges from every other node to it. Global state is useful for incorporating graph features to the decision making [9] and there is some evidence that it can facilitate logical reasoning [46]. Here, I will suppose that $x_0^{(\ell)}$ belongs to set $\mathcal{S}^{\gamma_\ell}$.

**Adaptive MPNN.** The forward-pass of an MPNN concludes after $d$ layers. However, the depth of a network may be adaptive [29–32]. In particular, $d$ may depend on the size and connectivity of the input graph or any adaptive computation time heuristic [29, 30] based on, for example, the convergence of the node representation [31, 32]. In the same spirit, in the following it will supposed that *all* hyper-parameters of an MPNN, such as its depth, width, message-size, and global state size, can be adaptively decided based on the input graph $G$.

## 2.2 Communication capacity

An MPNN $g$ can be thought of as a communication network $N(G, g)$, having processors as nodes and with connectivity determined by the input graph $G$. $N(G, g)$ operates in $\ell = 1, \ldots, d$ synchronous communication rounds and $m_\ell$ symbols are transmitted in round $\ell$ from each processor $v_i$ to each one its neighbors $v_j$ such that $e_{ij} \in \mathcal{E}$. Further, the processors have limited and round-dependent memory: in round $\ell$ the processors corresponding to nodes $\mathcal{V}$ can store $w_\ell$ symbols, whereas the external memory processor $v_0$ can store $\gamma_\ell$ symbols.

The communication complexity of a message-passing neural network corresponds to the maximum amount of information that can be sent in $N(G, g)$ between disjoint sets of nodes:

**Definition 2.1** (Communication capacity)**.** Let $g$ be an MPNN and fix a graph $G = (\mathcal{V}, \mathcal{E})$. For any two disjoint sets $\mathcal{V}_a, \mathcal{V}_b \subset \mathcal{V}$, the communication capacity $c_g$ of $g$ is the maximum number of symbols that $N(G, g)$ can transmit from $\mathcal{V}_a$ to $\mathcal{V}_b$ and from $\mathcal{V}_b$ to $\mathcal{V}_a$.

To understand Definition 2.1, imagine that the node-disjoint subgraphs $G_a = (\mathcal{V}_a, \mathcal{E}_a)$ and $G_b = (\mathcal{V}_b, \mathcal{E}_b)$ of $G$ are controlled by two parties: Alice and Bob (see Figure 1). In practice, Alice and Bob

correspond to two sub-networks of $g$. By construction, when Alice needs to send information to Bob, she does so by sending information across some paths that cross between $\mathcal{V}_a$ and $\mathcal{V}_b$. Bob does the same. From this elementary observation, it can be deduced that the number of symbols that can be sent during the forward pass is bounded by the cut between the two subgraphs:

**Lemma 2.1.** *Let $g$ be an MPNN of $d$ layers, where each has width $w_\ell$ (i.e., number of channels), exchanges messages of size $m_\ell$, and maintains a global state of size $\gamma_\ell$. For any disjoint partitioning of $\mathcal{V}$ into $\mathcal{V}_a$ and $\mathcal{V}_b$, the communication complexity of $g$ is at most*

$$c_g \leq cut(\mathcal{V}_a, \mathcal{V}_b) \sum_{\ell=1}^{d} \min\{m_\ell, w_\ell\} + \sum_{\ell=1}^{d} \gamma_\ell,$$

*with $cut(\mathcal{V}_a, \mathcal{V}_b)$ being the size of the smallest cut that separates $\mathcal{V}_a$ and $\mathcal{V}_b$ in $G$.*

Whenever the MPNN involves sending for each $e_{ij} \in \mathcal{E}$ two messages, i.e., one from $v_i$ to $v_j$ and one from $v_j$ to $v_i$, every edge should be counted twice in the calculation of $cut(\mathcal{V}_a, \mathcal{V}_b)$.

It is also interesting to remark that $c_g$ may be a random quantity. In particular, when $G$ is sampled from a distribution $\mathbb{D}$, the capacity of an *adaptive* MPNN, i.e., a network whose hyper-parameters change as a function the input, may vary as well. For this reason, the analysis will also consider the expected communication capacity $c_g(\mathbb{D})$ of $g$ w.r.t. $\mathbb{D}$.

## 2.3 Communication complexity

Let us momentarily diverge from graphs and suppose that Alice and Bob wish to jointly compute a function $f : \mathcal{X}_a \times \mathcal{X}_b \to \mathcal{Y}$ that depends on both their inputs. Alice's input is an element $x_a \in \mathcal{X}_a$ and Bob sees an element $x_b \in \mathcal{X}_b$. Later on, $x_a$ and $x_b$ will correspond to $G_a$ and $G_b$, respectively, whereas $y \in \mathcal{Y}$ will be the classification output (see Figure 1b).

To compute $f(x_a, x_b)$, the two parties need to exchange information based on some communication *protocol* $\pi$. Concretely, $\pi$ determines for each input $(x_a, x_b)$ the sequence $\pi(x_a, x_b) = ((\mathrm{ID}_1, s_1), (\mathrm{ID}_2, s_2), \ldots)$ of symbols that are exchanged, with each symbol $s_i \in \mathcal{S}$ being paired with the id of its sender (Alice or Bob)—for a more detailed description, see Appendix B. The number of symbols exchanged by $\pi$ to successfully compute $f(x_a, x_b)$ are denoted by $\|\pi(x_a, x_b)\|_m$, with subscript $m \in \{one, both\}$ indicating whether "successful computation" entails one or both parties figuring out $f(x_a, x_b)$ at the end of the exchange.

**Worst-case complexity.** The focus of classical theory is on the worst-case input. The *communication complexity* [33] of $f$ is defined as

$$c_f^m := \min_{\pi} \max_{(x_a, x_b) \in \mathcal{X}_a \times \mathcal{X}_b} \|\pi(x_a, x_b)\|_m \tag{1}$$

and corresponds to the minimum worst-case length of any protocol that computes $f$.

**Expected complexity.** In machine learning, one usually cares about the expected behavior of a learner when its input is sampled from a distribution. Concretely, let $(X_a, X_b)$ be random variables sampled from a distribution $\mathbb{D}$ with domain $\mathcal{X}_a \times \mathcal{X}_b$. The expected length of a protocol $\pi$ is

$$\mathrm{E}_{\mathbb{D}}\big[c_f^m(\pi)\big] := \sum_{(x_a, x_b) \in \mathcal{X}_a \times \mathcal{X}_b} \|\pi(x_a, x_b)\|_m \cdot \mathrm{P}(X_a = x_a, X_b = x_b), \tag{2}$$

where now the protocol length $\|\pi(x_a, x_b)\|_m$ is weighted according to the probability of each input. With this in place, I define the *expected communication complexity* of $f$ as

$$c_f^m(\mathbb{D}) := \min_{\pi} \mathrm{E}_{\mathbb{D}}\big[c_f^m(\pi)\big], \tag{3}$$

corresponding to the minimum expected length of any protocol that computes $f$.

For an overview of the classical theory of communication complexity pertaining to the worst-case and an analysis of the newly-defined expected complexity, the reader may refer to Appendix B.

To use communication complexity for learning problems $f : \mathcal{X} \to \mathcal{Y}$ from a graph universe $\mathcal{X}$ to a set of classes $\mathcal{Y}$ one needs to decompose (a subset of) $\mathcal{X}$ as $\mathcal{X}_a \times \mathcal{X}_b$. As it will be seen in the

following sections, the decomposition can be achieved by finding a disjoint partitioning of every graph $G \in \mathcal{X}$ into subgraphs $G_a \in \mathcal{X}_a$ and $G_b \in \mathcal{X}_b$, held by Alice and Bob, respectively. Then, in the worst case, $c_f^m$ symbols need to be exchanged so that one ($m=one$) or both ($m=both$) parties can correctly classify $G$ into class $y = f(G)$. Moreover, if $G$ is sampled from some distribution $\mathbb{D}$, then the two parties need to exchange at least $c_f^m(\mathbb{D})$ symbols in expectation. Together with Lemma 2.1, the aforementioned bounds can be used to characterize what an MPNN cannot achieve as a function of its worst-case and expected capacity.

# 3    Hardness results for determining the isomorphism class

This section derives necessary conditions for the communication capacity of a network that determines the graph isomorphism class of its inputs. This entails finding a mapping $f_{\text{isom}} : \mathcal{X} \to \mathcal{Y}$ from a universe of labeled graphs to their corresponding isomorphism classes. Crucially, though the nodes of graph $G$ are assigned some predefined order (which constitutes their label in graph-theory nomenclature), the class $f_{\text{isom}}(G)$ should be invariant to this ordering.

As it will be shown, MPNN of sub-quadratic and sub-linear capacity cannot compute the isomorphism class of connected graphs and trees, respectively:

**Theorem 3.1.** *Let $g$ be a MPNN using either a majority-voting or a consensus based readout (defined in Section 3.2). Denote by $c_g$ its communication capacity.*

1. *To compute $f_{isom}$ for every graph and tree of $n$ nodes, it must be that $c_g = \Omega\left(n^2\right)$ and $c_g = \Omega\left(n\right)$, respectively.*

2. *If each graph is sampled from $\mathbb{B}_{n/2,p}$ (defined in Theorem 3.3) to compute $f_{isom}$ in expectation it must be that $c_g(\mathbb{D}) = \Omega\left(n^2\right)$. Further, if each graph is a tree sampled from $\mathbb{T}_{n/2}$ (defined in Theorem 3.4) to compute $f_{isom}$ in expectation it must be that $c_g(\mathbb{D}) = \Omega\left(n\right)$.*

For general graphs, these results are one or two orders of magnitude tighter than arguments that compare the receptive field of a neural network with the graph diameter. Specifically, connected graphs have diameter at most $n$ and thus a diameter analysis yields $d = \Omega(n)$ without a global state and $d = \Omega(1)$ with one (as any two nodes are connected by a path passing through $v_0$).

The tree distribution was chosen purposefully to demonstrate that the bounds are also relevant for the anonymous case, when MPNN can also be analyzed by equivalence to the 1-WL test [3, 4]. For trees, the 1-WL test requires $n$ iterations because there exists a tree of diameter $n$. However, since MPNN is equivalent to 1-WL only when the former is built using injective aggregation functions (i.e., of unbounded width [3, 4, 47]), the equivalence does not imply a relevant lower bound on the width/message-size/global-state-size of MPNN. Further, the communication complexity analysis introduced here yields tighter results in expectation: it asserts that one needs $\Omega(n)$ capacity on average, even though the average tree in $\mathbb{T}_{n/2}$ has $O(\sqrt{n})$ diameter (and thus 1-WL would require $d = \Omega(\sqrt{n})$ in expectation).

**Graph isomorphism testing.** There is also a close relation between $f_{\text{isom}}$ and the graph isomorphism testing problem (see Figure 1c). Specifically, methods for isomorphism testing [3, 4, 6] that compare graphs $G$ and $G'$ by means of some invariant representation or embedding

$$g(G) = g(G') \quad \text{if and only if} \quad f_{\text{isom}}(G) = f_{\text{isom}}(G')$$

can be expressed as $g = q \circ f_{\text{isom}}$ for some injective function $q$. Since $q$ does not involve any exchange of information, the communication complexity of such testing methods is the same as that of $f_{\text{isom}}$. The proposed hardness results thus still hold.

The rest of this section is devoted to proving Theorem 3.1. The analysis consists of two parts: the communication complexity of distinguishing graphs and trees is derived in Section 3.1, and the implications of these results to MPNN are discussed in Section 3.2.

## 3.1    Communication complexity analysis

Rather than focusing directly on the universe of all graphs and trees, respectively, it will be convenient to analyze a strictly smaller universe $\mathcal{X}$ containing easily partitioned graphs. As it will be seen, we

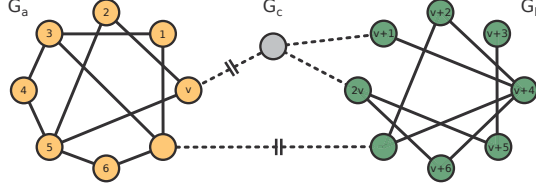

Figure 2: A visual depiction of a graph $G = (\mathcal{V}, \mathcal{E})$ chosen from $\mathcal{X}$. $G_a$ (in yellow) and $G_b$ (in green) are chosen from families $\mathcal{X}_a$ and $\mathcal{X}_b$ of graphs with $v$ nodes. The edges of $G_c$ (dashed lines) may connect to any node but should induce a $(\mathcal{V}_a, \mathcal{V}_b)$-cut of at most $\tau$.

can utilize such a restriction without significant loss of generality, because the derived worst-case impossibility results also apply to any universe that is a strict superset of $\mathcal{X}$.

Concretely, $\mathcal{X}$ will consist of all labeled graphs $G = (\mathcal{V}, \mathcal{E})$ on $n$ nodes admitting to the following $(\mathcal{X}_a, \mathcal{X}_b, \tau)$ decomposition:

    (a) Subgraph $G_a = (\mathcal{V}_a, \mathcal{E}_a)$ induced by labels $\mathcal{V}_a = (1, 2, \cdots, v)$ belongs to $\mathcal{X}_a$.

    (b) Subgraph $G_b = (\mathcal{V}_b, \mathcal{E}_b)$ induced by labels $\mathcal{V}_b = (v+1, v+2, \cdots, 2v)$ belongs to $\mathcal{X}_b$.

    (c) Subgraph $G_c = (\mathcal{V}, \mathcal{E} \setminus (\mathcal{E}_a \cup \mathcal{E}_b))$ yields $cut(\mathcal{V}_a, \mathcal{V}_b) \leq \tau$.

An example $(\mathcal{X}_a, \mathcal{X}_b, \tau)$ decomposable graph is depicted in Figure 2. This decomposition is fairly general: the main restriction placed is that the cut between $\mathcal{V}_a$ and $\mathcal{V}_b$ is bounded by $\tau$. Families $\mathcal{X}_a$ and $\mathcal{X}_b$ can be chosen to contain relevant families of graphs (e.g., all connected graphs or all trees), whereas $G_c$ may be selected arbitrarily. To derive lower bounds, it will be imagined that $G_a$ and $G_b$ are known by Alice and Bob, respectively, while both know $G_c$. The goal of the two parties is to determine $f_{\text{isom}}(G) = f_{\text{isom}}(G_a, G_b, G_c)$ by exchanging as little information as possible.

Two main results will be proven: Section 3.1.1 will show that, when $\mathcal{X}_a$ and $\mathcal{X}_b$ contain all labeled connected graphs on $v$ nodes, the worst-case and expected communication complexity are both $\Theta(v^2)$. Moreover, in Section 3.1.2 it is proven that, when $\mathcal{X}_a$ and $\mathcal{X}_b$ contain only trees, the two complexities are $\Theta(v)$.

### 3.1.1 Distinguishing connected graphs

When $\mathcal{X}_a$ and $\mathcal{X}_b$ contain all connected graphs on $v$ nodes, Alice and Bob should exchange $\Theta(v^2)$ symbols in the worst case:

**Theorem 3.2** (Worst-case complexity). *When $\mathcal{X}_a$ and $\mathcal{X}_b$ each contain the set of all connected graphs on $v$ nodes, the worst-case communication complexity of $f_{isom}$ is at least*

$$O(v^2) = c_{f_{isom}}^{both} \geq \frac{v^2}{\log_2 s} - 2v \log_s \left( \frac{v\sqrt{2}}{e} \right) - \log_s \left( 2ve^2 \right) + o(1) = \beta = \Omega \left( v^2 \right)$$

*and* $O(v^2) = c_{f_{isom}}^{one} \geq \frac{\beta - (\log_2 s)^{-1}}{2} = \Omega \left( v^2 \right).$

A similar bound holds also in the *random graph model* $\mathbb{G}_{v,p}$. In $\mathbb{G}_{v,p}$, every graph with $v$ nodes and $k$ edges is sampled with probability

$$\mathrm{P}(G \sim \mathbb{G}_{v,p}) = p^k (1-p)^{\binom{v}{2} - k}.$$

Effectively, this means the probability of choosing each graph depends only on the number of edges it contains. Moreover, for $p = 0.5$ each graph is sampled uniformly at random from the set of all possible graphs. The following theorem bounds the expected communication complexity when the subgraphs known to Alice and Bob are sampled from $\mathbb{G}_{v,p}$:

**Theorem 3.3** (Expected complexity). *Let $G_a$ and $G_b$ be sampled independently from $\mathbb{G}_{v,p}$, with $\log v / v < p < 1 - s^{\Omega(1)}$ and $cut(\mathcal{V}_a, \mathcal{V} \setminus \mathcal{V}_a) = cut(\mathcal{V}_b, \mathcal{V} \setminus \mathcal{V}_b) = 1$. Denote by $\mathbb{B}_{v,p}$ the resulting distribution. With high probability,*

$$O(v^2) = c_{f_{isom}}^{both} (\mathbb{B}_{v,p}) \geq v^2 \, \mathrm{H}_s(p) - v \left( 2 \log_s \left( \frac{v}{e} \right) + \mathrm{H}_s(p) \right) - \log_s \left( 2ve^2 \right) = \beta = \Omega(v^2)$$

*and*

$$O(v^2) = c^{one}_{f_{isom}}(\mathbb{B}_{v,p}) \geq \frac{\beta}{2} - \frac{v^2 - v(1 - \mathrm{H}_2(p)) + 1}{2\log_2 s} = \Omega(v^2),$$

*where* $\mathrm{H}_s(p) = -(1-p)\log_s(1-p) - p\log_s p$ *is the binary entropy function (base s).*

The expected complexity, therefore, grows asymptotically with $\Theta(v^2)$ and is maximized when every graph in the universe is sampled with equal probability, i.e., for $p = 0.5$. Interestingly, in this setting, the bounds of Theorems 3.2 and Theorem 3.3 match. This implies that, unless there is some strong isomorphism class imbalance in the dataset, the communication complexity lower bound posed by Theorem 3.2 does not only concern rare worst-case inputs, but should be met on average.

In the theorem it is asserted that $\log v/v < p < 1 - s^{\Omega(1)}$. The aforementioned lower bound suffices to guarantee that every $G \sim \mathbb{B}_{v,p}$ will be connected with high probability, whereas the upper bound is needed to ensure that $\mathrm{H}_s(p) = \Omega(1)$.

### 3.1.2 Distinguishing trees

Distinguishing trees (connected acyclic undirected graphs) is significantly easier:

**Theorem 3.4.** *Suppose that $G_a$ and $G_b$ are sampled independently from the set of all trees on $v$ nodes. Denote by $\mathbb{T}_v$ the resulting distribution. The communication complexity of $f_{isom}$ is at least*

$$O(v) = c^{both}_{f_{isom}} \geq c^{both}_{f_{isom}}(\mathbb{T}_v) \gtrsim 2v\log_s \alpha - 5\log_s v + \log_s 7 = \beta = \Omega(v)$$

*and* $O(v) = c^{one}_{f_{isom}} \geq c^{one}_{f_{isom}}(\mathbb{T}_v) \gtrsim \frac{\beta + \log_s 2}{2} = \Omega(v)$, *where* $\alpha \approx 2.9557652$ *and* $f(n) \gtrsim g(n)$ *means* $f(n) \geq g(n)$ *as n grows.*

Akin to the general case, the expected and worst-case complexities match when every tree is sampled with equal probability. Since a distribution over trees cannot be meaningfully parametrized based a connection probability $p$ (trees always have the same number of edges), by default in $\mathbb{T}_v$ every $G \in \mathcal{X}$ is sampled with equal probability.

### 3.2 Consequences for message-passing neural networks

Two types of networks are distinguished depending on how the readout function operates:

1. READOUT performs *majority-voting*. Specifically, for $g$ to compute $f_{isom}(G)$ there should exist a function $r : \mathcal{S}^{w_d} \to \mathcal{Y}$ and a set of nodes $\mathcal{M}_G \subseteq \mathcal{V}$ possibly dependent on $G$ and of cardinality at least $|\mathcal{M}_G| \geq \mu = O(1)$, such that $r(x_i^{(d)}) = f_{isom}(G)$ for every $v_i \in \mathcal{M}_G$.

2. READOUT performs *consensus*. This is akin to a majority-voting, with the distinction that $\mathcal{M}_G$ should contain at least $|\mathcal{M}_G| \geq n - \mu = \Omega(n)$ nodes.

The implications of a communication complexity bound to MPNN capacity are as follows:

**Lemma 3.1.** *Let $\mathbb{D}$ be a distribution over graphs that is supported on a universe $\mathcal{X}$ admitting to a $(\mathcal{X}_a, \mathcal{X}_b, \tau)$ decomposition. Further, suppose that $g$ is an MPNN whose communication capacity is always bounded from above by $c_g$ and is at most $c_g(\mathbb{D})$ in expectation. The following hold:*

1. *There exists some $G \in \mathcal{X}$ for which computing $f_{isom}(G)$ necessitates $c_g \geq c^m_{f_{isom}}$. In addition, for every $\mathcal{X}' \supset \mathcal{X}$ network $g$ cannot compute $f_{isom}(G)$ for some $G \in \mathcal{X}'$.*

2. *In expectation, computing $f_{isom}$ necessitates $c_g(\mathbb{D}) \geq c^m_{f_{isom}}(\mathbb{D})$. Moreover, if $c_g < \delta\, c^m_{f_{isom}}(\mathbb{D})$ for some $\delta \in [0, 1]$, then $g$ cannot compute $f_{isom}(G)$ with probability at least $(1 - \delta)/((\beta_m/c^m_{f_{isom}}(\mathbb{G})) - \delta)$.*

*Above, with majority-voting one should set $m = one$ and $v > (n - \mu)/2$, whereas with consensus $m = both$ and $v > \mu$. Further, $\beta_m$ is the worst-case length of a protocol with optimal expected length.*

With Lemma 3.1 in place, the proof of Theorem 3.1 follows from Theorems 3.2, 3.3 and 3.4 by setting $v = n/2$.

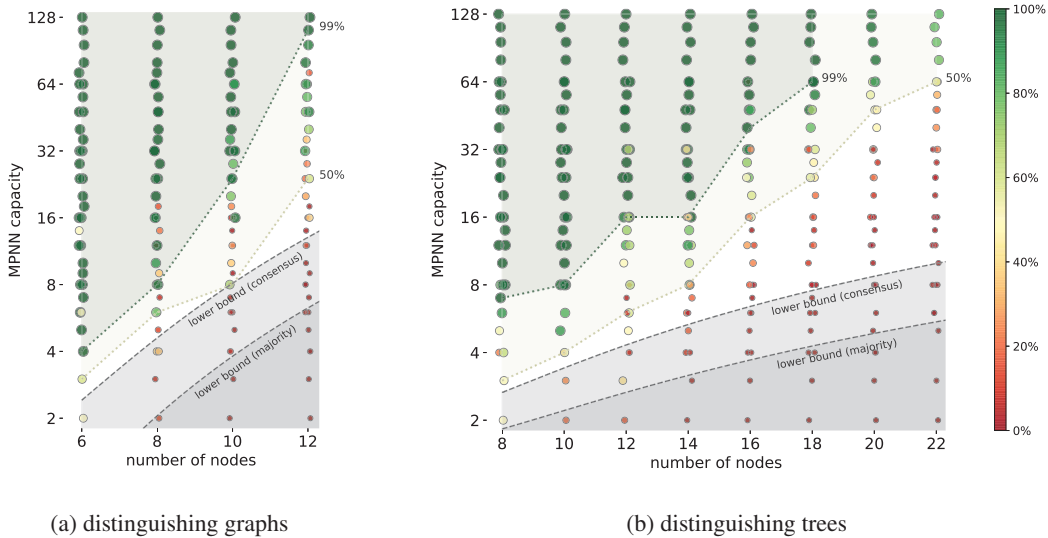

(a) distinguishing graphs          (b) distinguishing trees

Figure 3: Test accuracy in terms of communication capacity and the number of nodes for 4 graph (left) and 8 tree isomorphism tasks (right). Each marker corresponds to a trained network. Networks of high (low) accuracy as plotted with large green (small red) markers. The two dashed colored lines connect the smallest-capacity networks that attain 50% and 99% accuracy, respectively. The two gray regions at the bottom of the figure correspond to the proposed distribution-dependent lower bounds for a majority and consensus readout function. Best seen in color.

## 4 Empirical results

This section tests the developed theory on 12 graph and tree isomorphism classification tasks of varying difficulty. In the 420 neural networks tested, the bounds are found to consistently predict when each network can solve a given task as a function of its capacity.

### 4.1 Experimental setting

MPNN of different capacities were tasked with learning the mapping between a universe of graphs their corresponding isomorphism classes.

*Datasets.* A total of 12 universes were constructed following the theory: $\mathcal{X}_{\text{graph}}^n$ for $n = (6, 8, 10, 12)$ and $\mathcal{X}_{\text{tree}}^n$ for $n = (8, 10, \ldots, 22)$. Each $\mathcal{X}_{\text{graph}}^n$ was built in two steps: First, geng [48] was used to populate $\mathcal{X}_a$ and $\mathcal{X}_b$ with all possible connected graphs on $v = {}^n/_2$ nodes. Then, each $G \in \mathcal{X}_{\text{graph}}^n$ was generated by selecting $G_a$ and $G_b$ from $\mathcal{X}_a$ and $\mathcal{X}_b$ and connecting them with an edge, such that $\tau = 1$. The labels added to the nodes of $G$ were the one-hot encoding of a random permutation of $(1, \ldots, v)$ and $(v + 1, \ldots, n)$. The construction of $\mathcal{X}_{\text{tree}}^n$ differed only in that $\mathcal{X}_a$ and $\mathcal{X}_b$ contained all trees on $v = {}^n/_2$ nodes. Then, the 12 datasets were built by sampling graphs from each respective universe. These were split into a training, a validation, and a test set (covering 90%, 5%, and 5% of the dataset, respectively). Additional details are provided in Appendix A.

*Architecture and training.* The networks combined multiple GIN0 [3] layers with batch normalization and a simple sum readout. Their depth and width varied in $d \in (2, 3, 4, 5, 6, 7, 8)$ and $w \in (1, 2, 4, 8, 16)$, respectively, the message-size was set equal to $w$, and no global state was used. Each network was trained using Adam with a decaying learning rate. Early stopping was employed when the validation accuracy reached 100%.

### 4.2 Findings

Let me begin by stating that networks of sufficient size could solve nearly every task up to 100% test accuracy (Table 2 in Appendix A), which corroborates previous theoretical findings that non-

anonymous MPNN are universal and can solve graph isomorphism [23, 5], as well as that they can learn to be permutation invariant [22]. On the other hand, anonymous MPNN are always permutation equivariant but cannot distinguish between graphs of more than three nodes [6].

Figures 3a and 3b summarize the neural network performance for all the tasks considered. The achieved accuracy strongly correlated with communication capacity (computed based on Lemma 2.1) with larger-capacity networks performing consistently better. Moreover, in qualitative agreement with the analysis, solving a task can be seen to necessitate larger capacity when the number of nodes is increased. A case in point, whereas a capacity of 4 suffices to classify 99% of graphs of 6 nodes correctly, for 8, 10, and 12 nodes the required capacity increases to 8, 24, and 112, respectively. This identified correlation between capacity and accuracy could not be explained by the depth or width of the network alone, as, in most instances, tasks that could not be solved by wide and shallow networks could also not be solved by deep networks of the same capacity. The only exception was when receptive field did not cover the entire graph (see Figures 6a and 6b in Appendix A).

The gray regions at the bottom of each figure indicate the proposed expected communication complexity lower bounds. Here, $|\mathcal{S}| = 2$ based on the interpretation that each neuron can be either in an activated state or not. There are also two lower bounds plotted since a network that sums the final layer's node representations can learn to differentiably approximate both a majority-voting and a consensus function. The analysis asserts that a network with capacity below the gray dashed lines should not be able to correctly distinguish input graphs for a significant fraction of all inputs (see precise statement in Lemma 3.1). Indeed, networks in the gray region consistently perform poorly. The empirical accuracy appears to match closely the consensus bound, though it remains inconclusive if the network is actually learning to do consensus. A closer inspection (see Figures 5a and 5b in Appendix A) also reveals that the poor performance of networks in the gray region is not an issue of generalization. In agreement with the theory, networks of insufficient communication capacity do not possess the expressive power to map a fraction of all inputs to the right isomorphism class, irrespective of whether these graphs appear in the training or test set.

## 5    Conclusion

This work proposed a hardness-result for distinguishing graphs in the MPNN model by characterizing the amount of information the nodes can exchange during their forward pass (termed communication capacity). From a practical perspective, the results herein provide evidence that, if the amount of training data is not an issue, determining the isomorphism class of graphs is hard but not impossible for MPNN. Specifically, it was argued that the number of parameters needs to increase quadratically with the number of nodes. The implication is that, in the most general case, networks of practical size should be able to solve the problem for graphs with at most a few dozen nodes, but will encounter issues otherwise.

## Broader Impact

As we rely on neural networks more heavily, we are unfortunately sacrificing some of our ability to understand how our computers solve problems. Our lack of insight hinders us from using our technology to its full potential and can yield mistrust to the public. After all, if we cannot understand what a neural network is (capable of) doing, how can we know whether it is solving the correct problem? Poor understanding of fundamentals can also lead researchers to misguided optimism, believing that, given the right hyper-parameter tweaking and a large enough training set, neural networks can solve their problem. When incorrect, this mindset can lead to a waste of precious resources, such as time and energy.

In this light, impossibility results, such as those presented in this work, provide an insight into the fundamental limits of neural networks. Hardness results for graph neural networks, in particular, characterize the relational pattern recognition ability of practical networks and provide necessary conditions for using our tools to solve classical graph problems. The central implication of the results presented in this work is that one cannot expect to learn algorithms that distinguish (even approximately) connected graphs and trees unless the network size grows at-least polynomially with the graph size.

## Acknowledgments

I would like to express gratitude to the anonymous reviewers, as well as Nathanaël Perraudin, Nikolaos Karalias and Giovanni Cherubin for their insightful comments. I am also thankful to the Swiss National Science Foundation for financially supporting this work in the context of the project "*Deep Learning for Graph-Structured Data*" (grant number PZ00P2 179981).

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
