[Supplementary Material]

# A  Additional empirical results

This section presents the empirical results more comprehensively.

First, Table 1 provides summary statistics for each of the 12 tasks considered:

|  | $\mathcal{X}_{\text{graph}}^{6}$ | $\mathcal{X}_{\text{graph}}^{8}$ | $\mathcal{X}_{\text{graph}}^{10}$ | $\mathcal{X}_{\text{graph}}^{12}$ | $\mathcal{X}_{\text{tree}}^{8}$ | $\mathcal{X}_{\text{tree}}^{10}$ | $\mathcal{X}_{\text{tree}}^{12}$ | $\mathcal{X}_{\text{tree}}^{14}$ | $\mathcal{X}_{\text{tree}}^{16}$ | $\mathcal{X}_{\text{tree}}^{18}$ | $\mathcal{X}_{\text{tree}}^{20}$ | $\mathcal{X}_{\text{tree}}^{22}$ |
|---|---|---|---|---|---|---|---|---|---|---|---|---|
| classes | 3 | 21 | 231 | 6328 | 3 | 6 | 21 | 66 | 276 | 1128 | 5671 | 22730 |
| degree (avg.) | 4.0 | 4.7 | 5.4 | 6.0 | 3.5 | 3.6 | 3.7 | 3.7 | 3.8 | 3.8 | 3.8 | 3.8 |
| diameter (avg.) | 3.7 | 4.5 | 5.0 | 5.4 | 4.0 | 4.3 | 5.0 | 5.4 | 6.0 | 6.4 | 6.9 | 7.3 |
| dataset size | 10k | 10k | 40k | 100k | 10k | 10k | 40k | 40k | 40k | 40k | 40k | 100k |

Table 1: Details relevant to the 4 graph and 8 tree isomorphism tasks.

Some example graphs sampled are shown in Figure 4.

Figure 4: Example graphs sampled from two $(\mathcal{X}_a, \mathcal{X}_b, 1)$ decompositions. Top: $\mathcal{X}_a$ and $\mathcal{X}_b$ contain all connected graphs on $v = 6$ nodes (special case of Theorems 3.2 and 3.3). Bottom: $\mathcal{X}_a$ and $\mathcal{X}_b$ contain all trees on 11 nodes (special case of Theorem 3.4). In both cases, there exists a $\tau = 1$ cut between the nodes $\mathcal{V}_a$ controlled by Alice (in yellow) and nodes $\mathcal{V}_b$ controlled by Bob (in green).

Table 2 provides empirical evidence that, with a one-hot encoding of the node-ordering given as features and a sufficiently large training set, MPNN of sufficient capacity can solve graph isomorphism. In the current experiment, a large network (depth = 10 and width = 32) is seen to solve most isomorphism instances. The network did not achieve perfect classification for larger graphs, but better results can be achieved with more training data.

| accuracy | $\mathcal{X}_{\text{graph}}^{6}$ | $\mathcal{X}_{\text{graph}}^{8}$ | $\mathcal{X}_{\text{graph}}^{10}$ | $\mathcal{X}_{\text{graph}}^{12}$ | $\mathcal{X}_{\text{tree}}^{8}$ | $\mathcal{X}_{\text{tree}}^{10}$ | $\mathcal{X}_{\text{tree}}^{12}$ | $\mathcal{X}_{\text{tree}}^{14}$ | $\mathcal{X}_{\text{tree}}^{16}$ | $\mathcal{X}_{\text{tree}}^{18}$ | $\mathcal{X}_{\text{tree}}^{20}$ | $\mathcal{X}_{\text{tree}}^{22}$ |
|---|---|---|---|---|---|---|---|---|---|---|---|---|
| training | 100% | 100% | 100% | 99.997% | 100% | 100% | 100% | 100% | 100% | 100% | 100% | 100% |
| validation | 100% | 100% | 100% | 100% | 100% | 100% | 100% | 100% | 100% | 100% | 97.45% | 82.82% |
| test | 100% | 100% | 100% | 99.96% | 100% | 100% | 100% | 100% | 100% | 100% | 97.35% | 82.92% |

Table 2: The performance of a large-capacity MPNN.

The achieved accuracy of all networks considered is shown in Figures 5a and 5b for graph and tree isomorphism tasks, respectively. In contrast to the figures of Section 4, these plots depict the training as well as testing accuracy. For the majority of tasks the test and training accuracy is almost identical. Overfitting can be a problem for larger graphs (e.g., trees of at least 20 nodes). The problem can be mitigated by increasing the size of the training set.

Finally, Figures 6a and 6b demonstrate that depth and width are partially exchangeable. This implies that the correlation between capacity and accuracy (see Figures 3a and 3b) cannot be explained by only looking at the depth or width of a network. Here, the two figures depict the empirical test accuracy (by the marker color and size) as a function of depth and width for all tasks. For each task, the depth and width have been normalized by the square root of the critical capacity, corresponding to the smallest communication capacity of any network that could achieve at least 50% accuracy. As a consequence of the normalization, all networks in the top-right region (in white) possess sufficient capacity for the task at hand. Moreover, networks plotted below (above) the main diagonal are deeper

(a) graph isomorphism

(b) tree isomorphism

Figure 5: Training and test accuracy as a function of communication capacity.

than they are wide (wider than they are deep). As seen, the classification task can be solved by both wide and deep networks of super-critical capacity, as long as the networks are not too shallow. Indeed, networks of very small depth cannot see the entire graph and thus have poor accuracy.

(a) graph isomorphism

(b) tree isomorphism

Figure 6: Accuracy as a function of capacity-normalized depth and width. Depth and width are partially exchangeable for graph and tree isomorphism.

# B   Communication complexity: basics and beyond

## B.1   Basic theory: protocols

Let us start by denoting by $\mathcal{S}$ the common set of symbols[1] Alice and Bob use to communicate and denote by $s = |\mathcal{S}|$ its cardinality. A protocol $\pi$ is described in terms of a rooted $s$-ary tree, i.e., a tree with a clearly defined root and in which every internal node has exactly $s$ children. In addition,

every internal node $i$ is owned by either Alice or Bob and each one of the node's children symbolizes a symbol sent by its owner. Specifically, the protocol associates $i$ with a function $\pi_i$ that maps the input of $i$'s owner to $\mathcal{S}$ (or equivalently to one of $i$'s children). The protocol operates as follows: first, both parties set the *current* node to be the root of the tree. Say that the current node is $i$. If the owner of $i$ is Alice then she announces symbol $\pi_i(x)$ and otherwise Bob announces $\pi_i(y)$. Both parties then update the current node to point to the child of $i$ indicated by the value of $\pi_i$. This procedure is repeated until a leaf is found.

Figure 7: The execution of $\pi$ over input $(x_a, x_b)$ is a path within the $s$-ary tree. The decision of which symbol to send is taken by the node's owner (Alice or Bob) as a function of the current path and input. In this example there are $s = 2$ symbols $\mathcal{S} = \{0, 1\}$ and the path moves to the left/right child when the symbol 0/1 is sent. The protocol terminates at the leaf and the output is $\pi(x_a, x_b) = ((A, 0), (B, 1), (A, 1))$.

By definition, the number of symbols $\|\pi(x_a, x_b)\|_m$ Alice and Bob need to send in order to jointly compute $f(x_a, x_b)$ using protocol $\pi$ equals the length of the path from the root to the leaf $\pi(x_a, x_b)$. Moreover, the number of symbols sent by a protocol in the worst case (i.e., for any input) is at most equal to the depth of the protocol tree (Fact 1.1 in [33]).

## B.2 Basic theory: monochromatic rectangles

To understand how protocols operate one needs to consider the concept of rectangles. A *rectangle* is a subset of $\mathcal{X}_a \times \mathcal{X}_b$ that can be expressed as $\mathcal{X}'_a \times \mathcal{X}'_b$ for some $\mathcal{X}'_a \subset \mathcal{X}_a$ and $\mathcal{X}'_b \subset \mathcal{X}_b$. Intuitively, if one represents $\mathcal{X}_a \times \mathcal{X}_b$ as a matrix $\boldsymbol{X}$ of size $|\mathcal{X}_a| \times |\mathcal{X}_a|$, then a rectangle is any principal submatrix $\boldsymbol{X}'$ of $\boldsymbol{X}$, i.e., a matrix that contains a subset of rows and columns.

As it turns out, every protocol can be described in terms of rectangles. Let $\mathcal{R}^i \subseteq \mathcal{X}_a \times \mathcal{X}_b$ be the set of inputs leading to a path that crosses a node $i \in \pi$. Moreover, define the following sets:

$$\mathcal{X}_a^i = \{x \in \mathcal{X}_a : \exists y \in \mathcal{X}_b \text{ such that } (x_a, x_b) \in \mathcal{R}^i\}$$
$$\mathcal{X}_b^i = \{y \in \mathcal{X}_b : \exists x \in \mathcal{X}_a \text{ such that } (x_a, x_b) \in \mathcal{R}^i\}$$

The following result clarifies the connection between protocols and rectangles.

**Lemma B.1** (Lemma 1.4 in [33]). *For every protocol $\pi$ and node $i$, $\mathcal{R}^i$ is a rectangle with $\mathcal{R}^i = \mathcal{X}_a^i \times \mathcal{X}_b^i$. Further, the rectangles $\mathcal{R}^\ell$ given by the leafs $\ell \in \mathcal{L}_\pi$ of the protocol tree form a partition of $\mathcal{X}_a \times \mathcal{X}_b$.*

Effectively, at any point in a protocol, a rectangle describes the different possible outputs of $f$ given the messages that have been exchanged. Every new message that the two parties exchange, eliminates some possible outputs, decreasing the size of the rectangle.

With this in place, it is not hard to realize that, for every leaf $\ell \in \mathcal{L}_\pi$, the function $f$ should always take the same value at every $(x_a, x_b) \in \mathcal{R}^\ell$ in order for both parties to be able to compute the output from $\pi(x_a, x_b)$. Such rectangles are referred to as *monochromatic*: concretely, a rectangle $\mathcal{R} \subset \mathcal{X}_a \times \mathcal{X}_b$ is monochromatic if $f(x_a, x_b) = f(x'_a, x'_b)$ for every $(x_a, x_b), (x'_a, x'_b) \in \mathcal{R}$. Indeed, if leaf rectangles were not monochromatic, Alice and Bob would not be able to identify the output of $f$ based on $\mathcal{R}^\ell$.

The following theorem is obtained by combining Lemma B.1 with the fact that the minimum depth of any $s$-ary tree with $s^c$ leafs is $c$.

**Theorem B.1** (Theorem 1.6 by Rao and Yehudayoff [33]). *If the communication complexity of $f : \mathcal{X}_a \times \mathcal{X}_b \to \mathcal{Y}$ is $c_f^{both}$, then $\mathcal{X}_a \times \mathcal{X}_b$ can be partitioned into at most $s^{c_f^{both}}$ monochromatic rectangles.*

The following is a direct corollary:

**Corollary B.1** (Rao and Yehudayoff [33]). *If $\mathcal{X}_a \times \mathcal{X}_b$ cannot be partitioned into $s^c$ monochromatic rectangles, then $c_f^{both} \geq c$.*

A simple way to satisfy the requirement of the corollary is to prove that no large monochromatic rectangle exists. For instance, if it is shown that all monochromatic rectangles have size bounded by $k^2$ then every monochromatic partitioning must contain at least $|\mathcal{X}_a \times \mathcal{X}_b|/k^2$ rectangles and the complexity is at least $c_f^{both} \geq \log_s \left(|\mathcal{X}_a \times \mathcal{X}_b|/k^2\right)$. I will rely on this method in the following to derive lower bounds on the worst-case communication complexity of different functions.

## B.3 A different perspective: expected communication complexity

The following lemma connects the expected communication complexity $\mathrm{E}_{\mathbb{D}}[c_f(\pi)]$ to the entropy of the categorical distribution induced by the leafs of the protocol tree.

**Lemma B.2.** *Let the random variables $X = (X_a, X_b) \sim \mathbb{D}$ be sampled from some distribution $\mathbb{D}$ and, moreover, suppose that the random variable $L_\pi$ is the leaf for a protocol $\pi$ that computes $f(X_a, X_b)$. The expected communication complexity of $f$ is*

$$\min_\pi \mathrm{H}_s(L_\pi) \leq c_f^m(\mathbb{D}) \leq \min_\pi \mathrm{H}_s(L_\pi) + 1,$$

*where $\mathrm{H}_s(L_\pi)$ is the Shannon entropy (base $s$) of $L_\pi$ under $\mathbb{D}$.*

*Proof.* The expected length of a protocol $\pi$ is

$$\mathrm{E}_{\mathbb{D}}\left[c_f^m(\pi)\right] = \sum_{x_a, x_b} \|\pi(x_a, x_b)\|_m \cdot \mathrm{P}(X_a = x_a, X_b = x_b)$$

$$= \sum_{\ell \in \mathcal{L}_\pi} \mathrm{depth}(t) \cdot \mathrm{P}(L_\pi = \ell)$$

$$= \mathrm{E}_{\mathbb{D}}[\mathrm{depth}(L_\pi)].$$

Note that the set $\mathcal{L}_\pi$ contains the leafs of the protocol tree and $L_\pi$ is a categorical random variable over leafs with

$$\mathrm{P}(L_\pi = \ell) = \sum_{x,y \,:\, \pi(x_a, x_b) = \ell} \mathrm{P}(X_a = x_a, X_b = x_b),$$

which is also equal to the probability $\mathrm{P}\left((X_a, X_b) \in \mathcal{R}^\ell\right)$ that a randomly drawn input belongs to $\mathcal{R}^\ell$.

To understand $c_f^m(\mathbb{D})$ it helps to realize the connection between protocols and coding theory: rather than sending information between Alice and Bob, one may think of sending the leafs over a channel by using a codebook. In this analogy, each leaf corresponds to a code and the path from the root of the protocol tree to every internal node at depth $t$ corresponds to code prefix of length $t$. Furthermore, the probability of encountering the leaf is $P(L_\pi = t)$ and the depth of the protocol tree for every input $(x_a, x_b) \in \mathcal{R}^\ell$ is equal to the length of the code required to send the associated symbol.

From the above it follows that the act of designing a protocol with minimal $c_f^m(\pi)$ is equivalent to finding a tree with minimum expected path length from the root to the leafs. The latter is, in turn, equivalent to minimizing the length of the expected code length for a categorical distribution $L_\pi$. Therefore, based on Shannon's source coding theorem we have that

$$\min_\pi \mathrm{H}_s(L_\pi) \leq c_f^m(\mathbb{D}) \leq \min_\pi \mathrm{H}_s(L_\pi) + 1,$$

matching the lemma statement. □

(a) Example partitioning            (b) Maximum flow reduction

Figure 8: An example of the reduction employed in the proof of Lemma 2.1. The yellow and green subgraphs correspond respectively to $G_a$ and $G_b$. The global state (external memory) is shown in orange. Each edge is annotated based on its capacity in the maximum flow reduction.

## C  Deferred proofs

### C.1  Proof of Lemma 2.1

The number of symbols that can be transmitted from Alice to Bob in layer $\ell$ is bounded by the maximum flow of the following *multi-source multi-sink maximum flow problem with node capacities*:

- The nodes $\mathcal{V}_a$ are the senders and the nodes $\mathcal{V}_b$ are the sinks.
- Each edge has capacity $m_\ell$.
- Each node in $\mathcal{V}$ has capacity $w_\ell$, whereas $v_0$ has capacity $\gamma_\ell$.

This problem can be reduced to a simple maximum flow problem (single source single-sink without node capacities) in three steps:

1. All nodes in $\mathcal{V}_a$ (resp. $\mathcal{V}_b$) are connected to a new node $A$ (resp. $B$) with edges of infinite capacity.

2. Each node $v_i$ (with the exception of $A, B$ and $v_0$) is split into two nodes $\text{in}_i$ and $\text{out}_i$ connected by an edge of capacity $w_\ell$. Incoming edges to $v_i$ are connected to $\text{in}_i$ and outgoing edges are connected to $\text{out}_i$.

3. The same splitting procedure is performed for node $v_0$, but now the internal edge has capacity $\gamma_\ell$.

Consider the transformed flow network as shown in Figure 8b. By the max-flow min-cut theorem, the maximum value of the flow is equal to the minimum capacity over all cuts that separate $\mathcal{V}_a \cup A$ from $\mathcal{V}_b \cup B$. The latter however can always be bounded by $\text{cut}(A, B) + \gamma_\ell$. The first term of this equation gives the weight of the smallest cut separating $A$ and $B$ in the reduced graph, excluding those (orange) edges that touch $v_0$: since the edges from $A$ to $\mathcal{V}_a$ have infinite capacity (resp. from $B$ to $\mathcal{V}_b$), every such cut also separates $\mathcal{V}_a$ and $\mathcal{V}_b$. Notice also that every path from $A$ to $B$ includes at least one internal edge of capacity $w_\ell$ and one normal edge of capacity $m_\ell$. Combining the previous observations one finds that $\text{cut}(A, B) \leq \text{cut}(\mathcal{V}_a, \mathcal{V}_b) \min\{w_\ell, m_\ell\}$, where $\text{cut}(\mathcal{V}_a, \mathcal{V}_b)$ is the size of the smallest cut that separates $\mathcal{V}_a$ and $\mathcal{V}_b$ on $G$ (the undirected and unweighted graph prior to the reduction). The internal edge capacity of $v_0$ in accounted by term $\gamma_\ell$. The final expression is obtained by summing the bound over all $d$ layers.

### C.2  Proof of Theorem 3.2

The proof consists of two main steps. First, the number of monochromatic rectangles of $f_{\text{isom}}$ will be controlled using the number of graph isomorphism classes in $\mathcal{X}$. Then, invoking Corollary B.1 will result in a bound for $c_{f_{\text{isom}}}^{both}$. Second, the identified lower bound will be translated to a bound regarding $c_{f_{\text{isom}}}^{one}$ based on Lemma D.1.

There are $2^{\binom{v}{2}}$ labeled graphs on $v$ nodes (i.e., counting orderings), the overwhelming majority of which are connected. The number of connected labeled graphs on $v$ nodes is

$$|\mathcal{X}_a| = |\mathcal{X}_b| = 2^{\binom{v}{2}}\left(1 - \frac{2v}{2^v} + o\left(\frac{1}{2^v}\right)\right) = 2^{\binom{v}{2}}\left(1 - O\left(\frac{v}{2^v}\right)\right),$$

which, for sufficiently large $v$, is very close to $2^{\binom{v}{2}}$ [49, p. 138]. Specifically, one may write

$$\log_2 |\mathcal{X}_a| = \log_2 |\mathcal{X}_b| = \log_2\left(2^{\binom{v}{2}}\left(1 - O\left(\frac{v}{2^v}\right)\right)\right)$$

$$= \binom{v}{2}\log_2 2 + \log_2\left(1 - O\left(\frac{v}{2^v}\right)\right)$$

$$\geq \frac{v(v-1)}{2} - O\left(\frac{v}{2^v}\right) \qquad (\log(1-x) \geq -O(1)x \text{ for } x = o(1))$$

$$= \frac{v(v-1)}{2} + o(1)$$

and, similarly, $\log_2 |\mathcal{X}_a| = \log_2 |\mathcal{X}_b| \leq \frac{v(v-1)}{2}$. The number of permutations on $v$ nodes is $v!$, which implies that the number $c(v)$ of isomorphism classes of $v$-node graphs is bounded by

$$\log_2 c(v) \geq \log_2\left(\frac{|\mathcal{X}_a|}{v!}\right) \tag{4}$$

$$= \frac{v(v-1)}{2} - \log_2(v!) + o(1) \tag{5}$$

$$\geq \frac{v(v-1)}{2} - v\log_2\left(\frac{v}{e}\right) - \log_2\left(\sqrt{ve^2}\right) + o(1) \qquad (\text{since } x! \leq \sqrt{xe^2}\,(x/e)^x)$$

$$= \frac{v^2}{2} - v\log_2\left(\frac{v\sqrt{2}}{e}\right) - \log_2\left(\sqrt{ve^2}\right) + o(1) \tag{6}$$

By construction, $\mathcal{X}$ contains at least $c(v)(1 + c(v))/2$ classes. To obtain this bound, one assumes that there do not exist any classes that differ only w.r.t. $G_c$ and then notes that each unique class of $\mathcal{X}$ may be build either by gluing two distinct or identical classes on $v$ nodes (corresponding to graphs in $\mathcal{X}_a$ and $\mathcal{X}_b$). The bound then follows by counting all pairs of elements (there are $c(v)$ of those) with repetitions (e.g., for $\{a, b, c\}$ the set of possible pairs are $\{(aa), (ab), (ac), (bb), (bc), (cc)\}$).

The number of monochromatic rectangles of $f_{\text{isom}}$ is at least the number of classes and thus Corollary B.1 asserts:

$$c_{f_{\text{isom}}}^{both} \log_2 s = \log_2\left(\left\{\begin{array}{c}\text{minimum number of}\\ \text{monochromatic}\\ \text{rectangles}\end{array}\right\}\right)$$

$$\geq \log_2\left(\frac{c(v)(c(v) + 1)}{2}\right)$$

$$= 2\log_2 c(v) + \log_2\left(1 + \frac{1}{c(v)}\right) - 1 \geq 2\log_2 c(v) - 1 \tag{7}$$

Substituting (6) into (7) gives:

$$c_{f_{\text{isom}}}^{both} \log_2 s \geq v^2 - 2v\log_2\left(\frac{v\sqrt{2}}{e}\right) - 2\log_2\left(\sqrt{ve^2}\right) - 1 + o(1)$$

$$= v^2 - 2v\log_2\left(\frac{v\sqrt{2}}{e}\right) - \log_2\left(2ve^2\right) + o(1)$$

A bound on $c_{f_{\text{isom}}}^{one}$ can be derived with the help of Lemma D.1:

$$c_{f_{\text{isom}}}^{one} \log_2 s \geq c_{f_{\text{isom}}}^{both} \log_2 s - \max_{G_b, G_c} \log_s\left(|\{f(G_a, G_b, G_c) : G_a \in \mathcal{X}_a\}|\right) \log_2 s$$

$$= 2\log_2 c(v) - 1 - \log_2 c(v)$$

$$\geq \frac{v^2}{2} - v\log_2\left(\frac{v\sqrt{2}}{e}\right) - \log_2\left(2e\sqrt{v}\right) + o(1).$$

This lower bound derivation finishes by factoring $c_{f_{\text{isom}}}^{one}$ as a function of $c_{f_{\text{isom}}}^{both}$.

To conclude the proof, one notes the following elementary upper bound: to compute $f_{\text{isom}}(G)$, Bob and Alice can simply send their entire edge-sets to each other and proceed to compute $f(G_a, G_b, G_c)$ independently. Then, since the number of edges of a graph $v$ nodes are $|\mathcal{E}_a|, |\mathcal{E}_b| \leq v(v-1)/2$, it suffices to exchange $c_{f_{\text{isom}}} \leq v(v-1)/\log_2 s = O(v^2)$ symbols.

## C.3 Proof of Theorem 3.3

I will begin by proving a more general result. Specifically, it will be shown that the expected communication complexity is directly bounded by the entropy of the isomorphism class of a graph sampled from $\mathbb{G}$.

**Lemma C.1.** *The expected number of symbols that Alice and Bob need to exchange to jointly compute the isomorphism class $f_{isom}(G)$ of a graph sampled from $G = (G_a, G_b, G_c) \sim \mathbb{G}$ is at least*

$$c_{f_{isom}}^{both}(\mathbb{G}) \geq \min_{G_c} \mathrm{H}_s\left(f_{isom}(G)|G_c\right).$$

*Proof.* The first step is to condition the expected communication complexity on $G_c$:

$$
\begin{aligned}
c_{f_{\text{isom}}}(\mathbb{G}) &= \min_\pi \mathrm{E}_\mathbb{G}[c_{f_{\text{isom}}}(\pi)] \\
&= \min_\pi \sum_{G_c} \mathrm{P}(G_c)\,\mathrm{E}_\mathbb{G}[c_{f_{\text{isom}}}(\pi)|G_c] && \text{(due to the law of total expectation)} \\
&= \min_\pi \sum_{G_c} \mathrm{P}(G_c)\,\mathrm{E}_\mathbb{G}[c_{f_c}(\pi)] && \text{(by the definition } f_c(\cdot,\cdot) := f_{\text{isom}}(\cdot,\cdot,G_c)) \\
&\geq \sum_{G_c} \mathrm{P}(G_c)\min_\pi \mathrm{E}_\mathbb{G}[c_{f_c}(\pi)] \geq \min_{G_c} c_{f_c}(\mathbb{G}).
\end{aligned}
$$

Denote by $\mathcal{L}_\pi$ the set of leafs of a protocol $\pi$ that computes $f_c$ and by $L_\pi$ the random variable induced by the distribution $\mathbb{G}$ (for brevity, the conditioning on $G_c$ remains implicit in the following). We have that

$$\mathrm{H}_s(L_\pi) = \sum_{\ell \in \mathcal{L}_\pi} \mathrm{P}(L_\pi = \ell) \log_s\left(\frac{1}{\mathrm{P}(L_\pi = \ell)}\right). \tag{8}$$

Upon closer consideration, there are $|\mathcal{Y}|$ types of leafs such that $\mathcal{L}_\pi = \bigcup_{y=1}^{|\mathcal{Y}|} \mathcal{L}_{\pi,y}$, with each subset $\mathcal{L}_\pi^l$ containing all leafs for which the protocol outputs the graph isomorphism class $y$. From Lemma D.2 and because $\mathcal{L}_{\pi,1}, \ldots, \mathcal{L}_{\pi,|\mathcal{Y}|}$ form a partitioning of $\mathcal{L}_\pi$, we may write:

$$\mathrm{H}_s(L_\pi) \geq \sum_{y=1}^{|\mathcal{Y}|} \mathrm{P}(L_\pi \in \mathcal{L}_{\pi,y}) \log_s\left(\frac{1}{\mathrm{P}(L_\pi \in \mathcal{L}_{\pi,y})}\right).$$

The term $\mathrm{P}(L_\pi \in \mathcal{L}_{\pi,y})$ seen above corresponds to the probability that class $y$ will appear in our sample:

$$\mathrm{P}(L_\pi \in \mathcal{L}_{\pi,y}) = \mathrm{P}(f(G_a, G_b, G_c) = y)$$

therefore, $\min_\pi \mathrm{H}_s(L_\pi) \geq \mathrm{H}_s(f(G)|G_c)$ and the claim follows. $\square$

Coming back to the setting of the main theorem, denote by $k_y = |\mathcal{E}_a| + |\mathcal{E}_b|$ the number of edges of the graphs in class $y$ (disregarding the edges $\mathcal{E}_c$). For every $G_c$, we have that

$$\mathrm{P}(f_{\text{isom}}(G) = y \mid G_c) = i_c(v)\, p^{k_y}(1-p)^{2\binom{v}{2}-k_y} = i_c(v)\, p^{k_y}(1-p)^{v(v-1)-k_y}.$$

Term $i_c(v)$ corresponds to the size of the corresponding isomorphism class. Specifically, when $p$ is not too small and $\text{cut}(\mathcal{V}_a, \mathcal{V} \setminus \mathcal{V}_a) = \text{cut}(\mathcal{V}_b, \mathcal{V} \setminus \mathcal{V}_b) = 1$, it can be inferred that each isomorphism class in the universe contains at most $2(v!)^2$ labeled graphs. The remaining $n! - 2(v!)^2$ permutations yield isomorphic graphs with cut larger than one.

**Claim C.1.** *For any $\delta > 0$, $cut(\mathcal{V}_a, \mathcal{V} \setminus \mathcal{V}_a) = cut(\mathcal{V}_b, \mathcal{V} \setminus \mathcal{V}_b) = 1$, and $p \geq (\delta + \log_v)/v$, we have $i_c(v) \leq 2(v!)^2$ with probability at least $e^{-2e^{-\delta}} + o(1)$.*

*Proof.* To see this consider a labeled graph $G \in \mathcal{X}$ and let $G' = (\mathcal{V}', \mathcal{E}')$ be a second labeled graph that is isomorphic to $G$, induced by the label permutation $\mathcal{V}' = (\Pi(u) : u \in \mathcal{V})$. I claim that, if there exist $v_i, v_j \in \mathcal{V}_a$ for which $\Pi(v_i) \in \mathcal{V}_a$ and $\Pi(v_j) \in \mathcal{V}_b$, then $G' \notin \mathcal{X}$ (and the same holds if there exist $v_i, v_j \in \mathcal{V}_b$ for which $\Pi(v_i) \in \mathcal{V}_b$ and $\Pi(v_j) \in \mathcal{V}_b$).

The claim is proven by contradiction: suppose (for now) that $G_a$ and $G_b$ are connected. Then, for every set $\mathcal{S}$ of cardinality $v$ that is a strict subset of *both* $\mathcal{V}_a$ and $\mathcal{V}_b$ ($\mathcal{S}$ corresponds to the nodes with labels $(1, \cdots, v)$ in $G'$) the cut between $\mathcal{S}$ and its complement must be $\mathrm{cut}(\mathcal{S}, \mathcal{V} \setminus \mathcal{S}) = \sum_{v_i, v_j} \{v_i \in \mathcal{S}$ and $v_j \notin \mathcal{S}\} = \sum_{v_i, v_j} \{v_i \in \mathcal{S}$ and $v_j \in (\mathcal{V}_a \setminus \mathcal{S})\} + \sum_{v_i, v_j} \{v_i \in \mathcal{S}$ and $v_j \in (\mathcal{V}_b \setminus \mathcal{S})\} \geq 1 + 1$. The latter, however, is impossible as we have assumed that $\forall G' \in \mathcal{X}$, we must have $\mathrm{cut}(\mathcal{V}'_a, \mathcal{V}' \setminus \mathcal{V}'_a) = \mathrm{cut}(\mathcal{V}'_b, \mathcal{V}' \setminus \mathcal{V}'_b) = 1$. Therefore, the only valid permutations $\Pi$ are those that abide to either (a) if $v_i \in \mathcal{V}_a \to \Pi(v_i) \in \mathcal{V}_a$ and if $v_i \in \mathcal{V}_b \to \Pi(v_i) \in \mathcal{V}_b$ (there are $(v!)^2$ such permutations), or (b) if $v_i \in \mathcal{V}_a \to \Pi(v_i) \in \mathcal{V}_b$ and if $v_i \in \mathcal{V}_a \to \Pi(v_i) \in \mathcal{V}_a$ (there are $(v!)^2$ such permutations).

In the studied distribution, there is a non-zero probability that a disconnected graph appears. However, the probability is exponentially small when $p > \log v/v$. It is well known (see e.g., Theorem 4.1 by Frieze and Karoński [50]) that, for any $\delta > 0$ and $p = \frac{\delta + \log v}{v}$, a random graph on $v$ nodes is connected with probability

$$\mathrm{P}(G_a \text{ is connected}) = \mathrm{P}(G_b \text{ is connected}) = e^{-e^{-\delta}} + o(1)$$

and, by independence, $\mathrm{P}(G \text{ is connected}) = e^{-2e^{-\delta}} + o(1)$. $\square$

Based on the above observation, the conditional entropy of $f(G)$ can be rewritten as

$$\mathrm{H}_2(f_{\mathrm{isom}}(G)|G_c) = \sum_{y \in \mathcal{Y}} \mathrm{P}(f_{\mathrm{isom}}(G) = y|G_c) \log_2 \left( \frac{1}{\mathrm{P}(f_{\mathrm{isom}}(G) = y|G_c)} \right)$$

$$\geq \sum_{k=0}^{v(v-1)} \frac{\binom{v(v-1)}{k}}{i_c(v)} i_c(v) \, p^k (1-p)^{v(v-1)-k_y} \log_2 \left( \frac{1}{i_c(v) \, p^k (1-p)^{v(v-1)-k}} \right)$$

$$= \sum_{k=0}^{v(v-1)} \binom{v(v-1)}{k} p^k (1-p)^{v(v-1)-k} \left( -\log_2 i_c(v) + v(v-1) \log_2 \left( \frac{1}{1-p} \right) + k \log_2 \left( \frac{1-p}{p} \right) \right)$$

$$= \log_2 \left( \frac{1-p}{p} \right) \left( \sum_{k=0}^{v(v-1)} \binom{v(v-1)}{k} p^k (1-p)^{v(v-1)-k} k \right) + v(v-1) \log_2 \left( \frac{1}{1-p} \right) - \log_2 i_c(v)$$

Let $B$ be a binomial random variable with parameters $v(v-1)$ and $p$. The summation term is equivalent to the expectation of $B$:

$$\sum_{m=0}^{v(v-1)} \binom{v(v-1)}{k} p^k (1-p)^{v(v-1)-k} k = \mathrm{E}[B] = v(v-1)p$$

and, therefore,

$$\mathrm{H}_2(L_\pi) \geq \log_2 \left( \frac{1-p}{p} \right) v(v-1)p + v(v-1) \log_2 \left( \frac{1}{1-p} \right) - \log_2 i_c(v)$$

$$= v(v-1)\mathrm{H}_2(p) - \log_2 i_c(v) \quad \text{(by definition } \mathrm{H}_2(p) = \log_2 \left( \frac{1-p}{p} \right) p + \log_2 \left( \frac{1}{1-p} \right))$$

$$= v(v-1)\mathrm{H}_2(p) - 2\log_2 v! - 1 \quad \text{(see Claim C.1 } i_c(v) \leq 2(v!)^2)$$

$$\geq v(v-1)\mathrm{H}_2(p) - 2 \left( v \log_2 \left( \frac{v}{e} \right) + \frac{1}{2} \log_2 \left( ve^2 \right) \right) - 1 \quad \text{(since } x! \leq \sqrt{xe^2} \, (x/e)^x)$$

$$= v^2 \mathrm{H}_2(p) - v \left( 2 \log_2 \left( \frac{v}{e} \right) + \mathrm{H}_2(p) \right) - \log_2 \left( 2ve^2 \right)$$

Invoking Lemma C.1, one obtains:

$$c_{f_{\text{isom}}}^{both}(\mathbb{B}_{v,p}) \geq \min_{G_c} \frac{\text{H}_2(f_{\text{isom}}(G)|G_c)}{\log_2 s}$$

$$\geq v^2\,\text{H}_s(p) - v\left(2\log_s\left(\frac{v}{e}\right) + \text{H}_s(p)\right) - \log_s\left(2ve^2\right) = \beta \qquad (9)$$

Then Lemma D.1 gives:

$$c_{f_{\text{isom}}}^{one}(\mathbb{B}_{v,p})\log_2 s \geq c_{f_{\text{isom}}}^{both}(\mathbb{B}_{v,p})\log_2 s - \max_{G_b,G_c}\log_s\left(|\{f_{\text{isom}}(G_a,G_b,G_c)\ :\ G_a \in \mathcal{X}_a\}|\right)\log_2 s$$

$$= c_{f_{\text{isom}}}^{both}(\mathbb{B}_{v,p})\log_2 s - \log_2\left(\frac{|\mathcal{X}_a|}{v!}\right)$$

$$= c_{f_{\text{isom}}}^{both}(\mathbb{B}_{v,p})\log_2 s - \frac{v(v-1)}{2} + \log_2(v!)$$

$$\geq v(v-1)\left(\text{H}_2(p) - \frac{1}{2}\right) - \left(v\log_2\left(\frac{v}{e}\right) + \frac{1}{2}\log_2\left(ve^2\right)\right) - 1$$

$$= v^2\text{H}_2(p) - \frac{v}{2}\left(2\log_2\left(\frac{v}{e}\right) + \text{H}_2(p)\right) - \frac{1}{2}\log_2\left(2ve^2\right) - \frac{v^2 - v + v\text{H}_2(p) + 1}{2}$$

$$= \frac{\beta\log_2 s - v^2 + v(1 - \text{H}_2(p)) - 1}{2}$$

implying $c_{f_{\text{isom}}}^{one}(\mathbb{B}_{v,p}) \geq \frac{\beta}{2} - \frac{v^2 - v(1 - \text{H}_2(p)) + 1}{2\log_2 s}$.

## C.4 Proof of Theorem 3.4

According to Otter [51], the number of unlabeled trees on $v$ nodes grows like

$$t(v) \sim c\,\alpha^v\,v^{-5/2},$$

where the values $c$ and $\alpha$ known to be approximately 0.5349496 and 2.9557652 (sequence A051491 in the OEIS). Moreover, it was shown in the proof of Theorem 3.2, the number of monochromatic rectangles is at least $(t(v) + 1)\,t(v)/2$.

Corollary B.1 then implies

$$c_{f_{\text{isom}}}^{both} \geq \log_s\left(\frac{(t(v) + 1)\,t(v)}{2}\right)$$

$$\geq \log_s\left(\frac{t(v)^2}{2}\right)$$

$$\sim 2\log_s\left(\alpha^v\,v^{-5/2}\right) - \log_s\left(c^2/2\right) \sim 2v\log_s\alpha - 5\log_s v + \log_s 7 = \beta$$

Further, from Lemma D.1 one can derive:

$$c_{f_{\text{isom}}}^{one} \geq c_{f_{\text{isom}}}^{both} - \max_{G_b,G_c}\log_s\left(|\{f(G_a,G_b,G_c)\ :\ G_a \in \mathcal{X}_a\}|\right)$$

$$= c_{f_{\text{isom}}}^{both} - \log_s t(v)$$

$$\sim \log_s\left(\alpha^v\,v^{-5/2}\right) - \log_s\left(c/2\right) \sim v\log_s\alpha - \frac{5}{2}\log_s v + \frac{1}{2}\log_s 14$$

implying $c_{f_{\text{isom}}}^{one} \geq \frac{\beta + \log_s 2}{2}$.

Let me now consider the case that $G$ is sampled uniformly at random from the set of all trees in $\mathcal{X}$. It is a consequence of Lemma C.1 that when the graph $(G_a, G_b, G_c) \sim \mathbb{G}$ (conditioned on $G_c$) is sampled uniformly at random from a collection of isomorphism classes, the expected communication complexity is at least

$$c_{f_{\text{isom}}}^{both}(\mathbb{T}_v) \geq \min_{G_c}\log_s|\{f(G_a,G_b,G_c)\ :\ G \in \mathcal{X} \text{ s.t. } G_c\}|.$$

This can be seed to be identical to the worst-case bound encountered above. The derivation thus can be carried out analogously (and the same holds for $c_{f_{\text{isom}}}^{one}(\mathbb{T}_v)$ by Lemma D.1).

Finally, the upper bound $O(v)$ follows by the same argument as in the proof of Theorem 3.2, where now the number of edges of each of $G_a$ and $G_b$ is $v - 1$.

## C.5 Proof of Lemma 3.1

In general terms, the impossibility statement comes as a consequence of the definition of communication complexity: if the number of required exchanged symbols exceeds the symbols the learner can exchange (i.e., its communication capacity) then the latter will not be able to identify exactly $f_{\text{isom}}$.

The specifics depend on the appropriate definition:

Majority-voting necessitates $|\mathcal{M}_G| \geq \mu$, meaning that when $|\mathcal{M}_G| \geq \mu > n - 2v$ at least one of the two parties should have gathered sufficient information to determine $f_{\text{isom}}(G)$ at the final layer. Therefore, $m$ should be "one". With consensus on the other hand, we have that $|\mathcal{M}_G| \geq n - \mu > n - v$ which implies that both parties need to know the class.

The worst-case communication complexity definition guarantees that there exists at least one input for which the required number of symbols is $c_{f_{\text{isom}}}^{(m)}$. Thus, since $\mathbb{D}$ is densely supported on $\mathcal{X}$, the impossibility must occur with strictly positive probability. The impossibility also applies to any universe $\mathcal{X}'$ that is a strict superset of $\mathcal{X}$. This can be easily derived by conditioning on $\mathcal{X} \subset \mathcal{X}'$ (which can only decrease the communication complexity) and repeating the analysis identically.

The implications of the expected complexity bound, are two-fold:

First, if $g$ is adaptive, its capacity $c_g$ is a random variable over the input distribution. The bound then asserts that $\mathrm{E}[c_g] \geq c_{f_{\text{isom}}}^m(\mathbb{D})$.

For networks of fixed size, one may derive a bound on the probability of error. Specifically, fix $\pi^*$ to be the protocol that achieves minimal expected length and let $\beta_m$ be an upper bound of $\pi^*$ length over all inputs. By Lemma D.3, for any $\delta \in [0,1]$ one has

$$\mathrm{P}\left(\|\pi^*(G)\|_m > \delta\, c_{f_{\text{isom}}}^m(\mathbb{D})\right) \geq \frac{1-\delta}{(\beta_m/c_{f_{\text{isom}}}^m(\mathbb{D})) - \delta}.$$

The above is a bound on the probability of error for a network that satisfies $c_g \leq 2\delta\, c_{f_{\text{isom}}}^m(\mathbb{D})$.

One can also generalize the previous result to distributions $\mathbb{D}'$ defined on a strict superset $\mathcal{X}'$ of $\mathcal{X}$ that is (up to normalization) identical with $\mathbb{D}$ within $\mathcal{X}$:

$$\text{for all } G \in \mathcal{X}: \quad \mathrm{P}(G \sim \mathbb{D}') = c\,\mathrm{P}(G \sim \mathbb{D}) \quad \text{with} \quad c = \sum_{G \in X} \mathrm{P}(G \sim \mathbb{D}').$$

Then, the probability of error w.r.t. $\mathbb{D}'$ is at least $c\,\frac{1-\delta}{(\beta_m/c_{f_{\text{isom}}}^m(\mathbb{D})) - \delta}$.

## D Helpful lemmata

**Lemma D.1.** *In the universe considered in Section 3, the following hold for any $\mathbb{D}$:*

$$c_f^{one} \geq c_f^{both} - \max_{G_b, G_c} \log_s\left(|\{f(G_a, G_b, G_c) \,:\, G_a \in \mathcal{X}_a\}|\right)$$

$$c_f^{one}(\mathbb{D}) \geq c_f^{both}(\mathbb{D}) - \max_{G_b, G_c} \log_s\left(|\{f(G_a, G_b, G_c) \,:\, G_a \in \mathcal{X}_a\}|\right).$$

*Proof.* Consider the setting of $c_f^{one}$, where for a successful termination it suffices for one party to compute the output of $f$. Suppose w.l.o.g., that this party is Alice. In particular, Alice determines class $y = f(G_a, G_b, G_c)$ based on a protocol $\pi$ of minimal length. In this setting, Bob does not know $y$ but he is aware of $\mathcal{X}_b^\ell$ (and $G_c$), where $\ell$ is the leaf of the protocol tree at input $(G_a, G_b, G_c)$. Therefore, both parties know that the class must belong to the set $\{f(G_a, G_b, G_c) \,:\, G_b \in \mathcal{X}_b^\ell \text{ and } G_a \in \mathcal{X}_a\}$. It is a consequence that there exists a protocol $\pi'$ of length

$$\|\pi'(G_a, G_b, G_c)\|_{both} \leq \|\pi(G_a, G_b, G_c)\|_{one} + \log_s |\{f(G_a, G_b, G_c) \,:\, G_b \in \mathcal{X}_b^\ell \text{ and } G_a \in \mathcal{X}_a\}|$$

that results in both parties knowing $y$. The protocol $\pi'$ entails first simulating $\pi$ and then Alice sending to Bob the index of $y$ in the set of feasible classes. Moreover, since $f$ corresponds to the graph isomorphism problem, for Alice to know $y$, she must also know the isomorphism class of Bob.

Therefore, the feasible set of classes contains only the feasible subgraph isomorphism classes of $G_a$, which are at most

$$|\{f(G_a, G_b, G_c) \,:\, G_b \in \mathcal{X}_b^{\ell} \text{ and } G_a \in \mathcal{X}_a\}| \leq \max_{G_b, G_c} |\{f(G_a, G_b, G_c) \,:\, G_a \in \mathcal{X}_a\}|$$

The claimed inequalities then follow by the optimality of the protocol $\pi$ and since the same construction can be repeated for every input. $\square$

**Lemma D.2.** *Let $X$ be a categorical random variable with sample space $\mathcal{X}$. For any partitioning $\mathcal{X} = \mathcal{A}_1, \cdots, \mathcal{A}_k$ we have that*

$$\mathrm{H}_s(X) \geq \sum_{i=1}^{k} \mathrm{P}(X \in \mathcal{A}_i) \log_s \left( \frac{1}{\mathrm{P}(X \in \mathcal{A}_i)} \right)$$

*Proof.* The proof is elementary. It relies on the inequality $\mathrm{P}(X = x) \leq \mathrm{P}(X \in \mathcal{A}_i)$ that holds for all $x \in \mathcal{A}_i$:

$$\mathrm{H}_2(X) = \sum_{i=1}^{k} \mathrm{P}(X \in \mathcal{A}_i) \sum_{x \in \mathcal{A}_i} \frac{\mathrm{P}(X = x)}{\mathrm{P}(X \in \mathcal{A}_i)} \log_s \left( \frac{1}{\mathrm{P}(X = x)} \right)$$

$$\geq \sum_{i=1}^{k} \mathrm{P}(X \in \mathcal{A}_i) \min_{x \in \mathcal{A}_i} \log_s \left( \frac{1}{\mathrm{P}(X = x)} \right)$$

$$= \sum_{i=1}^{k} \mathrm{P}(X \in \mathcal{A}_i) \log_s \left( \frac{1}{\max_{x \in \mathcal{A}_i} \mathrm{P}(X = x)} \right)$$

$$\geq \sum_{i=1}^{k} \mathrm{P}(X \in \mathcal{A}_i) \log_s \left( \frac{1}{\sum_{x \in \mathcal{A}_i} \mathrm{P}(X = x)} \right) = \sum_{i=1}^{k} \mathrm{P}(X \in \mathcal{A}_i) \log_s \left( \frac{1}{\mathrm{P}(X \in \mathcal{A}_i)} \right),$$

as claimed. $\square$

**Lemma D.3.** *For any random variable $X \leq \beta$ and $\delta \in [0, 1]$ we have $\mathrm{P}(X > \delta \, \mathrm{E}[X]) \geq \frac{1-\delta}{r-\delta}$, where $r = \beta/\mathrm{E}[X]$.*

*Proof.* For any $t \leq \beta$,

$$\mathrm{E}[X] = \sum_{x \leq t} \mathrm{P}(X) \, x + \sum_{x > t} \mathrm{P}(X) \, x \leq \mathrm{P}(X \leq t) \, t + \mathrm{P}(X > t) \, \beta = (1 - \mathrm{P}(X > t)) t + \mathrm{P}(X > t) \, \beta$$

or, equivalently, $\mathrm{P}(X > t) \geq (\mathrm{E}[X] - t)/(\beta - t)$. The final inequality is obtained by setting $t = \delta \mathrm{E}[X]$. $\square$

## Footnotes

[1]Though usually it is assumed that the parties communicate using binary symbols, i.e., $\mathcal{S} = \{0, 1\}$, the set could also be defined more abstractly to contain $s$ symbols.