[Reviews · NeurIPS 2020]

Review 1

Summary and Contributions: The paper deals with supervised graph classification via graph neural networks (GNNs)/MPNNs from a theoretical viewpoint. The paper introduces the notion of communication capacity/complexity (CC) of an MPNN, which quantifies the number of symbols/bits two subgraphs can exchange during the forward computation. The formalism is then used to study the ability of GNNs to determine/learn the isomorphism type a graph. Results: - upper bound on the CC via card. of cuts, the width of GNNs, and size of messages - introduction of worst-case complexity and expected complexity (graphs are sampled from a fixed but unknown distribution) - bound on the worst-case and expected C. of determining the iso. type of connected graphs and trees The paper is rounded off with (small-scale) controlled experiments on synthetic graphs verifying the usefulness of the theory in "practical" settings.

Strengths: - introduces a new framework for studying the limits of graph neural networks that gives some insights of the current limits of GNNs/MPNNs, different way of looking at the expressivity of GNNs - might spark further work

Weaknesses: - studying the learnability of the isomorphism type seems only of theoretical interest - studying the learnability of the iso. type of trees seems not very interesting as the 1-WL can already do that, hence also the (anonymous) GIN-eps layer - experiments seem somewhat "constructed" in a way to fit the theory

Correctness: Results seems plausible, however I did not check the proofs in detail.

Clarity: Clear writing, easy to follow.

Relation to Prior Work: Yes, the relation to related work is clearly stated. Discussion of related work is sufficient.

Reproducibility: Yes

Additional Feedback: Questions: - Are your results applicable to anonymous MPNNs, too? Is it agnostic to the the setting? - l. 71, do the results, e.g., Prop. 3.1, only hold for distributions over finite many graphs? - l. 91, what is a "non-Lipschitz" computer? - Section 3.3, How are your results connected to the fact that 1-WL at most needs v iterations to determine the iso. type of a tree? - Sec. 4.1, what is the reason for setting v = n/2 and \tau = 1? Do the emp. results also hold for sparser graphs and a bigger cut value? - l. 244, choosing w <= 16 does not agree with current practices. What happens if you increase w further? - Can your results be lifted to other graph properties beyond determining the iso. type? Suggestions: - make the distinction between communication capacity and complexity more clear Post rebuttal: ========== The rebuttal partly addressed some of my concerns. I worked I worked through the paper again, and I am somewhat torn here. The paper provides an innovative and interesting theoretical perspective on the limits of MPNNs. However, many parts lack clarity, even Definition 2.1 is not rigorous/lacks formality, i.e., it is not clear what is assumed for the MPNN. It merely captures an intuition in a non-formal way. Hence, I am not willing to raise my score.


Review 2

Summary and Contributions: This paper provides a theoretical study of message-passing graph neural networks (MPNNs). It focuses on non-anonymous MPNNs that allow vertices to be assigned identity - such MPNNs were studied in literature and shown to be universal, with the caveat that the ability to respect permutations (equivariance) needs to be learned (which is in contrast to the more standard graph neural neural networks). The author claims that the results presented in the paper are relevant both to anonymous and non-anonymous MPNNs Specifically, this paper focuses on understanding the complexity of MPNNs needed to correctly distinguish isomorphic graphs (connected graphs and trees, in particular). For that the author (I assume it is one author rather than multiple authors because of the use of singular throughout the paper, but I do not know the identity of the author, this review is still double-blind) uses tools from communication complexity. First the author defines communication capacity of an MPNN (this is specific to each single graph and depends on its properties). The author then upper-bounds communication capacity of MPNNs using cuts in the underlying graph. He then moves on to deriving lower bounds on communication complexity of distinguishing graphs (he focuses on connected graphs and trees). The author then gives bounds for the minimum capacity a network needs to have in order to correctly distinguish isomorphism classes. Finally, the author validates the conclusions of the paper experimentally. The paper is accompanied with a supplementary material which contains (among other) the proofs of all the results in the main text.

Strengths: The paper proposes an original and very reasonable approach to better understand the discriminative power of GNNs. It is well written and illuminating. The theory seems sound (but I am no expert in TCS, so I cannot guarantee I have not missed something). The proposed view based on communication complexity is also more fine-grained than the previous attempts to understand the strength of GNNs based purely on the product of their width and depth (which was done in recent papers).

Weaknesses: There is one claim in the paper that I particularly did not understand (which may be purely problem on my side, nonetheless it would be good if the author could clarify this). First, I did not understand precisely how the theoretical analysis applies to the case of anonymous MPNNs. As far as I (mis?)understood the argument, it is based on finding a lower bound for communicating enough information to distinguish isomorphism classes but anonymous MPNNs are only as powerful as Weisfeiler-Lehman algorithm (which for instance cannot distinguish non-isomorphic k-regular graphs from each other). So it is not clear to me what the value of the lower bound is for anonymous MPNNs. ----- AFTER REBUTTAL: The author has clarified this issue in his response. I am fine with the response. I agree that the lower bound for the anonymous case makes sense for trees. However, in case this paper gets in, the author should clarify this point in the final version of the paper.

Correctness: I tried to check all the proofs. However, I had absolutely no prior exposure to communication complexity therefore I might have easily overlooked some problems. I did not find any obvious problems.

Clarity: The paper reads well.

Relation to Prior Work: Correct. The only place where I was wondering whether existing work is not missing was the part about expected communication complexity. As I mentioned above, I know nothing about communication complexity (except what I read to be able to understand the present paper). My question is: Do people in communication complexity really not use standard coding theory?

Reproducibility: Yes

Additional Feedback: Page 3: What is non-Lipschitz computer in “…(and non-Lipschitz) computer…”? Page 3: “sent to from node” Page 4: “it be proven”
 Appendix between lines 530 to 531: From where did the second factor 2 appear? The same place as above: You say: ”log(1 − x) ≥ −O(1)x for x ∈ [0, 1])“. I do not think this is technically correct. It only holds on smaller subintervals of [0,1] if I understand what you are saying here. Anyways you need this statement to be true only for some interval [0,epsilon], I guess so this should not be a problem.


Review 3

Summary and Contributions: This paper offers several hardness results for MPNN in computing isomorphism classes of graphs. The paper shows that the communication capacity of an MPNN needs to be at-least linear in the size of the graph for learning isomorphism class of trees and quadratic for general graphs. The communication capacity is defined as the amount of symbols that can be transmitted between disjoint subgraphs of a given graph during the feedforward of the MPNN. It is bounded from above by the min-cut of the graph times channel and message size. The hardness results are derived using the above mentioned bound and a lower bound on the computational complexity of the isomorphism type function. That is, the amount of symbols that need to transferred between two disjoint subgraphs of a graph G to compute the isomorphism type of G. The main contribution of this paper is in the context of non-anonymous MPNN expressive power, since anonymous MPNNs are not even universal (as the authors acknowledge). The analysis method is novel, as far as I know, and uses interesting ideas of communication capacity and complexity. The main limitations of this paper are: exposition clarity and quality is lacking, the final result is not directly interpretable in terms of the MPNN architecture so it is not clear how it can be used to improve MPNN architectures, and it is not clear to what extent the main result here significantly improves upon [33]. Lastly, the computational complexity bounds feel a bit over complicated (see questions on this below).

Strengths: The anonymous MPNN regime is an active one due to its expressive power and some indications of good practical behavior. Therefore analyzing this model is of interest to the NeurIPS community. The notions of communication capacity and complexity are interesting ideas and could be proven useful additions to the mathematical GNN analysis toolbox.

Weaknesses: The exposition issues are detailed below. Other concerns I have are the following. The final result asserts that if the communication capacity bound is lower than the lower bound for f_iso then there exists a graph that N (the MPNN in question) cannot compute. I cannot extract a concrete intuition or conclusion regarding the MPNN architecture that improves upon the polynomial of the depth and width as in [33]. This is mainly due to the fact that the bound in Lemma 2.1 is defined for a SPECIFIC graph. When we consider ALL graphs (that include those with min-cut=1), it basically reduces to depth times width. Therefore, using this result in the context of MPNN for general class of graphs does not seem to significantly improve upon the observation of [33] that the depth times width should be at least polynomial in graph size. I understand f_iso is a different task than in [33] and has its independent merit, but i nevertheless think it's important to place this result in the correct context of previous work. I also have some questions regarding the communication complexity bounds in Theorems 3.1 and 3.2. The notion of protocol is not clear (also in the appendix the definition was not clear to me) and its instantiation in the MPNN case was obscure. Furthermore, since a graph with v vertices and e edges requires at-least Theta(e) symbols for representing it, and trees and general graphs have linear and quadratic edges at most, respectively, then the communication complexity has to be at-least that to compute the isomorphism class. Can't this argument be used instead of the rectangles arguments, which are not explained with sufficient clarity in the appendix?

Correctness: The main claims are justified in the appendix, and are provided only partially, relying on works of Rao and Yehudayoff making them hard to check, at least for me.

Clarity: I find the exposition to be a significant limitation of this paper. I feel that this paper can be of better value to the community by improving its exposition. I want to emphasis three aspects where I find the exposition lacking. First, the paper's goals as detailed in section 1.2 are not clear: what do you mean by worst case distributions (also in this paper the final result deals with existence of a single graph G, see Proposition 3.1)? What do you mean by general phenomenon? Making the goals of this paper clearer would improve it I believe. Second, the main technical part of the paper builds upon the ideas of communication capacity and complexity. These are not defined with sufficient rigor. The notion of protocol seems to be central to these concepts. It is not described at the main paper and it is not sufficiently clear in my opinion in the appendix. Furthermore, I would expect some informal proof or intuition to the lower bounds to be included in the main paper, after all this is a theoretical paper. Lastly, the main result of the paper is not formulated precisely (for example as a theorem) and only mentioned in passing at the end of section 3.4. I would expect implications to architectures of MPNN to be discussed and relations to the previous work [33] clearly indicated.

Relation to Prior Work: See above.

Reproducibility: Yes

Additional Feedback: Other comments: - Line 112: define the disjoint partitioning with respect to mathcal{V}. - What does "transmit" mean mathematically? - Lemma 2.1: shouldn't d be d/2 or something similar in the bound? - Line 157: f(x_a,x_b) means f_iso(G) of what G? I don't see how f_iso(G) is defined solely by x_a and x_b. UPDATE AFTER REBUTTAL: Thanks for providing a rebuttal. I think the suggestions for edits are acceptable and would improve the paper. I still didn’t fully understand the relation to [33] and the necessity of the rectangle analysis, and I hope it will be clearer in the next/final version. I raised my score accordingly.


Review 4

Summary and Contributions: This paper shows theoretical results on the analysis of message passing neural network (MPNN) for graph isomorphism problem. The main results are that the MPNN network capacity needs to grow linearly with the number of nodes for tree graphs and quadratically for general connected graphs.

Strengths: + This paper proposes an interesting new perspective of theoretically analyzing the power of MPNN through communication capacity. Theoretical results are established for one important problem - the graph isomorphism and might be useful for further analysis of MPNN in more general domains. + Discussions of the special case further strengthen the contribution + The paper is in general well-written.

Weaknesses: - The full derivation seems to be defined over one specific task, a walk through in figure 1 containing the MPNN connected with the graph example might be helpful for readers to understand the paper more clearly. - The definition of network capacity seems to be missing and is used interchangeably with network parameters.

Correctness: Overall looks good. Didn't fully check the derivation and proof.

Clarity: yes

Relation to Prior Work: yes

Reproducibility: Yes

Additional Feedback:

[Author Response · NeurIPS 2020]

First off, I would like to thank the reviewers for their helpful feedback. The reviewers agree that this work provides a novel way of looking at the expressivity of GNNs and establishes insightful results for the important problem of graph isomorphism. Following their suggestions, the paper's exposition will be improved by: 1) providing further (graphical) intuition on the concept of a protocol; 2) unifying Theorems 3.1-3.3 and Prop. 3.1 into a main theorem; and 3) by expliciting the differences with [33] and the 1-WL test (discussed below). I will also address any other minor comments that are not discussed here due to space limitations.

**R1: Why is the learnability of the isomorphism class ($f_{iso}$) important?** 1) $f_{iso}$ is a good proxy for graph classification: due to MLP universality, a GNN that solves $f_{iso}$ is sufficiently powerful to solve any graph classification problem on the same graph distribution (i.e., irrespective of how the classes are assigned). 2) The bounds also apply to any GI testing method (like [31]) that compares graphs by means of some invariant representation (see Sec.3:129-133). GI testing is a subject of intense study within the GNN community [11, 22, 31, 25].

**R1 & R2: Anonymity, and improvement over 1-WL bounds for trees.** The proposed bounds apply *both* to anonymous and non-anonymous MPNN. The tree distribution was chosen purposefully to demonstrate that the bounds are also relevant for the anonymous case. As R1/R2 mentioned, it is known that 1-WL can recognize $v$-node trees in $v$ iterations, simply because there exists a tree of diameter $v$. However, since MPNN is equivalent to 1-WL only when the former is built using injective aggregation functions (i.e., of unbounded width), the equivalence does not imply a relevant lower bound on the width/message-size/global-state-size of MPNN. Further, the communication complexity theoretic bound is tighter and more refined: e.g., it asserts that one needs $\Omega(v)$ capacity *in expectation*, even though the average tree has $O(\sqrt{v})$ diameter (and thus 1-WL would require depth=$\Omega(\sqrt{v})$ in expectation).

**R1: Experiments: why were they set-up in this manner and are they small-scale?** The experiments were intended as a verification of the bounds and thus were constructed to closely match the studied setting. Yet, since the considered distributions form a (size-able) subset of all possible graphs/trees on $n$ nodes, the demonstrated impossibility results will similarly hold for the full distribution. The hyper-parameters $v = n/2, \tau = 1, w \leq 16$ were selected to illustrate the bounds for small $n$—the bounds also hold for different settings of $v, \tau, w$, but then a larger $n$ would be needed to demonstrate the dependency (rendering the experiments more lengthy). Overall, the experiments considered ~36.5k different graphs and 420 different MPNN, lasting more than 672 GPU hours. In terms of these metrics, I would not consider them small-scale.

**R2: Communication complexity (CC) and coding theory.** The applicability of Shannon's theorem (proof of Lemma B.2) arises when shifting from a worst-case complexity definition (studied in CC) to an expected one (defined here). I believe that this is the reason why these connections were not exploited previously.

**R3: Relation to [33].** There are four main differences (beyond that [33] did not consider graph isomorphism): 1) *The current paper derives necessity bounds for the expected as well as worst case.* The results of [33] assert that there exists a distribution such that for *some* graph in the distribution $dw$ needs to depend on $n$ — however, the MPNN could still attain 99.99% accuracy without needing to satisfy the $dw$ condition. Differently, the current paper bounds the probability of error over the distribution (see third bullet in Prop. 3.1). Note also that Lemma 2.1 can be used to define a capacity bound for any arbitrary family of graphs (though the bound is tighter when the graphs in question can be jointly partitioned with a cut of at most $\tau$). Further, since anonymous MPNN is oblivious to node ordering, in the anonymous setting the bound is valid as long as each graph can be cut in two pieces of roughly equal size. 2) *In contrast to [33], the bounds here consider the message-size and apply to MPNN with global-state.* 3) *This paper considers graph classification (with readout); [33] considered node classification problems (no readout function).* 4) *Finally, the current paper derives lower bounds by developing a new, technically more involved, and insightful connection to CC.*

**R3: Regarding rigor and completeness.** 1) *All results were fully proven.* Employing a previous result (Corollary B1) in a proof is standard practice in theoretical papers and does not affect rigor. To help the reader, relevant CC arguments and results are summarized in App. B.1 and B.2. These might not suffice to provide a complete intuition, but they suffice for completeness. To aid the reader further, the camera-ready will include a more in-depth explanation of protocols and communication complexity. 2) The intuition behind the technical constructions is provided by the introduction of Section 3. Further intuition in the main text would require an understanding of App. B, which I believe goes beyond the interests of the casual reader.

**R3: Why is a rectangle-based analysis necessary?** The argument of the reviewer could be used in a setting where the information needs to be transmitted one-way. However, in MPNN any two subgraphs (parties) arrive to the output by exchanging information over multiple steps/layers (i.e., the transmission is both ways). This renders the approach suggested by the reviewer inapplicable and motivates the need for rectangles/communication complexity: a rectangle represents the uncertainty inherent to each party after every step of the communication exchange (protocol).

**R4: Communication capacity.** The definition can be found in Definition 2.1 (lines 112-114).

[Meta-Review · NeurIPS 2020]

This paper draws a connection between GNNs and the communication complexity theory, and derived some interesting results regarding the GNN size and its capability of solving certain classes of problems, on certain input data distributions. Given that this paper brings in communication complexity theory into the study of GNNs, which is mostly new to our reviewers, it is a quite demanding paper to thoroughly review, and the reviewer confidence is low because of the lack of context / time to check every claim. The AC carefully checked the proofs of all the major claims and believes that the paper is correct, and the techniques developed in this paper can potentially bring useful insights to the community and have broader impact. However the review process did show that the author should do a better job of making this paper more accessible to the machine learning community, by e.g. improving the clarity of various definitions that set up the theory (the definition of the communication protocol is particularly confusing, even though it is almost copied from [38]), and explaining the intuitive proof strategy of the main results.